# COMMUNICATIONS
nature

# MYC drives aggressive prostate cancer by disrupting transcriptional pause release at androgen receptor targets

Xintao Qiu [1,2,22], Nadia Boufaied[3,22], Tarek Hallal[3,4], Avery Feit [1,2], Anna de Polo[3,5], Adrienne M. Luoma [6], Walaa Alahmadi[3,7], Janie Larocque[3,7], Giorgia Zadra[8,9], Yingtian Xie[1,2], Shengqing Gu[1,2,10], Qin Tang[1,2,10], Yi Zhang [1,10], Sudeepa Syamala [1], Ji-Heui Seo[2], Connor Bell[2], Edward O'Connor[2], Yang Liu[11], Edward M. Schaeffer [12], R. Jeffrey Karnes[13], Sheila Weinmann[14], Elai Davicioni[11], Colm Morrissey[15], Paloma Cejas [1,2], Leigh Ellis [16,17,18], Massimo Loda [19], Kai W. Wucherpfennig [6], Mark M. Pomerantz[2], Daniel E. Spratt [20], Eva Corey [15], Matthew L. Freedman[1,2,21], X. Shirley Liu [1,10], Myles Brown [1,2], Henry W. Long [1,2,23✉] & David P. Labbé [3,4,5,7,23✉]

c-MYC (MYC) is a major driver of prostate cancer tumorigenesis and progression. Although MYC is overexpressed in both early and metastatic disease and associated with poor survival, its impact on prostate transcriptional reprogramming remains elusive. We demonstrate that MYC overexpression significantly diminishes the androgen receptor (AR) transcriptional program (the set of genes directly targeted by the AR protein) in luminal prostate cells without altering AR expression. Analyses of clinical specimens reveal that concurrent low AR and high MYC transcriptional programs accelerate prostate cancer progression toward a metastatic, castration-resistant disease. Data integration of single-cell transcriptomics together with ChIP-seq uncover an increase in RNA polymerase II (Pol II) promoter-proximal pausing at AR-dependent genes following MYC overexpression without an accompanying deactivation of AR-bound enhancers. Altogether, our findings suggest that MYC overexpression antagonizes the canonical AR transcriptional program and contributes to prostate tumor initiation and progression by disrupting transcriptional pause release at AR-regulated genes.

A full list of author affiliations appears at the end of the paper.

Prostate cancer is the most common non-cutaneous malignancy and a leading cause of cancer-related lethality in men[1]. The androgen receptor (AR), a ligand-activated transcription factor, is central to the homeostasis of normal prostate epithelium[2,3]. Importantly, since the discovery that prostate cancer is reliant on androgen signaling to thrive[4,5], targeting AR activity continues to be the main pillar of prostate cancer therapy[6].

Prostate cancer initiation and progression involves the corruption of the normal prostate cancer transcriptional network[7]. Loss of the *NKX3-1* homeobox gene is a frequent and early event in prostate cancer etiology while the *TMPRSS2-ERG* gene fusion and *FOXA1* mutations both identify major molecular subtypes of the disease[8,9].

Overexpression of c-Myc (MYC), a master transcription factor and oncoprotein whose expression and function are tightly controlled under normal circumstances, is frequently observed in prostate cancer. Nuclear overexpression of MYC protein is an early event observed in luminal cells of prostate intraepithelial neoplasia (PIN) and is maintained in a large proportion of primary carcinomas and metastatic disease[10]. Importantly, about 25% of familial risk of prostate cancer map to germline variation at chromosome 8q24 with mechanistic evidence tying this region to *MYC* regulation[11–13]. Critically, MYC overexpression in normal luminal cells of murine prostate is sufficient to initiate prostate cancer[14], providing evidence that deregulation of MYC protein expression is a critical oncogenic event driving prostate cancer initiation.

Although AR and MYC are both central to prostate cancer etiology, our current understanding of the interplay between these two transcription factors is scarce. A recent study revealed that MYC overexpression antagonizes androgen-induced gene expression in an androgen-sensitive cell line representative of advanced prostate cancer[15]. However, it remains unknown how increased MYC expression shapes the AR transcriptional program in normal luminal prostate cells as they transition to PIN and subsequently progress from a localized to a metastatic disease.

Here we model MYC-driven prostate cancer initiation in vivo and define the transcriptional rewiring occurring in luminal cells at a single-cell level. We demonstrate that MYC overexpression diminishes the canonical AR transcriptional program, alters the AR cistrome, and results in the establishment of a corrupted AR transcriptional program in a murine model. We determine that an active MYC transcriptional program and low AR activity identify prostate cancer patients predisposed to fail standard-of-care therapies and most likely to develop metastatic castration resistant prostate cancer (mCRPC). Accordingly, we find that high *MYC* mRNA expression in castration-resistant tumors is also associated with a weakened canonical AR transcriptional program and a repurposing of the AR cistrome. Patients harboring a mCRPC characterized by an active MYC transcriptional program and low AR activity are more likely to fail first-line next generation AR signaling inhibitor (ARSI; *i.e.* abiraterone acetate or enzalutamide) and die of their disease. Critically, integration of transcriptomic and epigenomic data reveals that MYC overexpression does not lead to the deactivation of AR-bound enhancers but instead results in RNA polymerase II (Pol II) promoter-proximal pausing at AR-dependent genes. Altogether, our findings suggest that MYC overexpression contributes to tumor initiation and progression by disrupting the AR transcriptional program.

## Results

**MYC induces a profound transcriptional reprogramming in murine prostate lobes**. To examine the transcriptional reprogramming associated with MYC-driven prostate cancer initiation,

we compared a 12-week-old mouse that overexpresses an $ARR_2Pb$ driven human *c-MYC* transgene (MYC) in the prostate epithelium to a wild-type (WT) littermate[14]. At 12 weeks of age, MYC overexpression induces cellular epithelium transformation to PIN, a premalignant condition that often precedes the development of invasive adenocarcinoma in humans[16], with varying penetrance across prostate lobes. Notably, the murine anterior prostate (AP) remained mostly unaffected by MYC overexpression while PIN penetrance reached 83% and 97% in the dorsolateral prostate (DLP) and ventral prostate (VP), respectively[17]. Transcriptional profiling of whole prostate lobes at a single-cell level revealed a strong overlap with the matched bulk gene expression profiling across lobes and genotypes (WT and MYC; Fig. 1a, b and Supplementary Fig. 1a). Comparison of gene expression levels quantified by single-cell RNA-seq (scRNA-seq; aggregate expression) or bulk RNA-seq revealed that scRNA-seq quantitatively recapitulates bulk gene expression (Fig. 1c and Supplementary Fig. 1b). Accordingly, with the exception of the AP, unsupervised clustering revealed a strong correlation between single-cell transcriptome and the matched bulk transcriptome (Fig. 1d) and revealed that MYC induces a profound transcriptional reprogramming in both the DLP and VP lobes (Fig. 1e).

**Single-cell transcriptome delineates inter- and intra-prostate lobe heterogeneity**. To determine key differences between murine prostate lobes, we projected the single-cell transcriptome data into the t-distributed stochastic neighbor embedding (tSNE) space. Using known markers (Supplementary Fig. 2a, b), we identified nine major subpopulations of cells across prostate lobes (Fig. 1f). Notably, basal cells ($Krt5^+$, $Krt14^{Hi}$) were the most abundant epithelial cell subtype observed in the AP and DLP lobes, whereas luminal cells ($Krt8^{Hi}$, $Krt18^{Hi}$) were overwhelmingly represented in the VP lobe. While murine *Myc* (mm10Myc) was expressed across all subpopulations and prostate lobes (Supplementary Figs. 2c and 3), human *c-MYC* transgene expression (hg19MYC) was largely restricted to the luminal subpopulation (Fig. 1g) and more prevalent in the VP lobe (Fig. 1h), a feature in line with the greater penetrance of the MYC-driven PIN transformation observed in the VP lobe (Fig. 1i)[17].

The high representation of luminal cells coupled with a robust and uniform MYC-driven PIN transition in the VP enabled us to further define distinct luminal subpopulations. K-means clustering revealed a luminal subpopulation ($Krt8^{Hi}$, $Krt18^{Hi}$) common to both WT and MYC genotypes and characterized by high expression of *Krt4* but negative for *Nkx3-1* expression ($Krt4^{Hi}$, *Nkx3-1*$^-$; Fig. 2a, b and Supplementary Fig. 4a). Concurrent high expression of *Cd44*, *Tacstd2* (Trop2) and *Psca* suggests that this subpopulation corresponds to luminal progenitor cells[18]. In untransformed VP, the main luminal cell cluster was composed of two subpopulations characterized by either high or low expression of androgen-responsive genes such as *Pbsn* and *Msmb* (Supplementary Fig. 4b)[19,20]. Human *MYC* was predominately expressed in luminal cells (Fig. 2c, d), resulting in an extensive transcriptional reprogramming within the luminal compartment (Fig. 2a, b). Importantly, the distinct transcriptional profile of human *MYC* overexpressing luminal cells was identifiable even without inclusion of the human *MYC* transcript in the generation of the tSNE plot (Supplementary Fig. 5). In agreement with MYC function in controlling transcriptional programs that favor cell growth and proliferation[21], we identified a subset of highly proliferative human *MYC* overexpressing luminal cells positive for cyclin B1, DNA topoisomerase II alpha and the marker of proliferation Ki-67 ($Ccnb1^+$, $Top2a^+$, $Mki67^+$; Fig. 2b and Supplementary Fig. 4c), a state that was independent of human or murine *MYC* transcript levels (Fig. 2d). Finally, a limited number

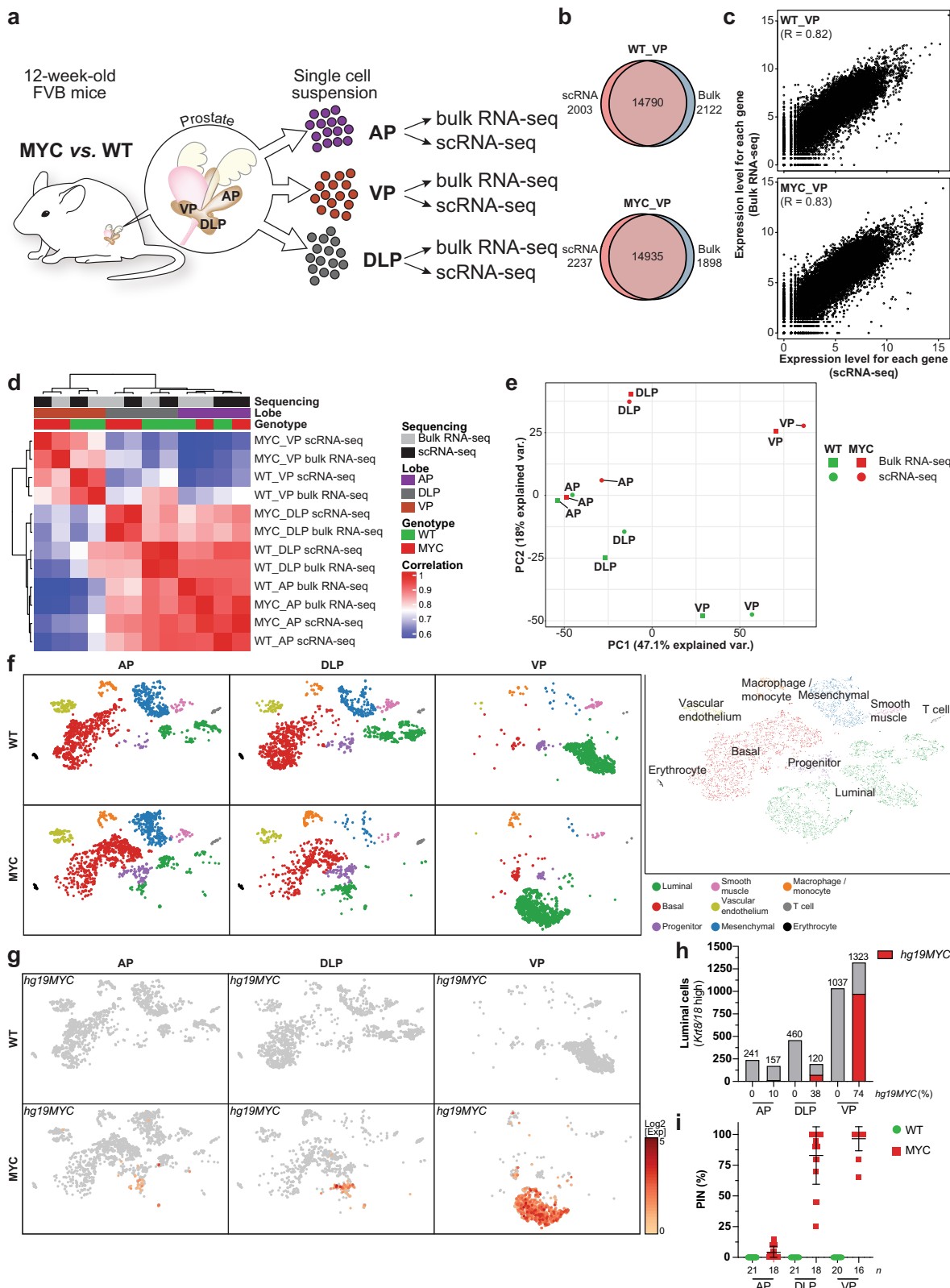

of cells belonging to hematopoietic ($Ptprc^+$), vascular endothelium ($Pdgfra^+$), smooth muscle ($Actg2^+$) and adipocyte ($Fabp4^+$) populations were also identified (Fig. 2b and Supplementary Fig. 4d). Taken together, these results demonstrate that MYC-driven transcriptional reprogramming can be readily captured in vivo by single-cell transcriptomics to expose inter- and intra-prostate lobe heterogeneity.

**MYC-driven luminal cells transformation dampens the AR transcriptional program**. To define the transcriptional reprogramming driven by *MYC* overexpression in the VP lobe across cell subpopulations, we created a pseudobulk sample for each subpopulation and performed Gene Sets Enrichment Analyses (GSEA) using the Hallmark gene sets[22]. As expected, the pseudobulk RNA-seq analysis showed that the MYC-driven

**Fig. 1 MYC induces a profound transcriptional reprogramming in murine prostate lobes. a** Graphical summary of the experimental design. **b**, **c** Transcriptional profiling of WT and MYC-transformed VP reveal high concordance for the total number of genes detected (**b**) and their expression levels (**c**) between bulk and single-cell RNA-seq (VP; matched bulk and single-cell RNA-seq; $n = 1$ per genotype). **d**, **e** Sample-sample correlation (**d**) and principal component analysis (**e**) between bulk and matched single-cell transcriptome identifies distinct transcriptional profiles across murine prostate lobes (AP, DLP, VP; matched bulk and single-cell RNA-seq; $n = 1$ per genotype). **f** Single-cell census of WT and MYC-transformed AP, DLP and VP. tSNE of scRNA-seq profiles colored using known markers identified nine major subpopulations across prostate lobes (AP, DLP, VP; $n = 1$ per genotype). **g–i** The human *MYC* transgene (*hg19MYC*) expression is largely restricted to the luminal compartment (**g** AP, DLP, VP; $n = 1$ per genotype) and predominantly expressed in the VP (**h** Source data are provided as a Source Data file), in accordance with the penetrance of prostatic intraepithelial neoplasia (**i** PIN; $n =$ biologically independent animals; mean ± SD; Source data are provided as a Source Data file[17]). WT: wild-type; VP: ventral prostate; DLP: dorsolateral prostate; AP: anterior prostate.

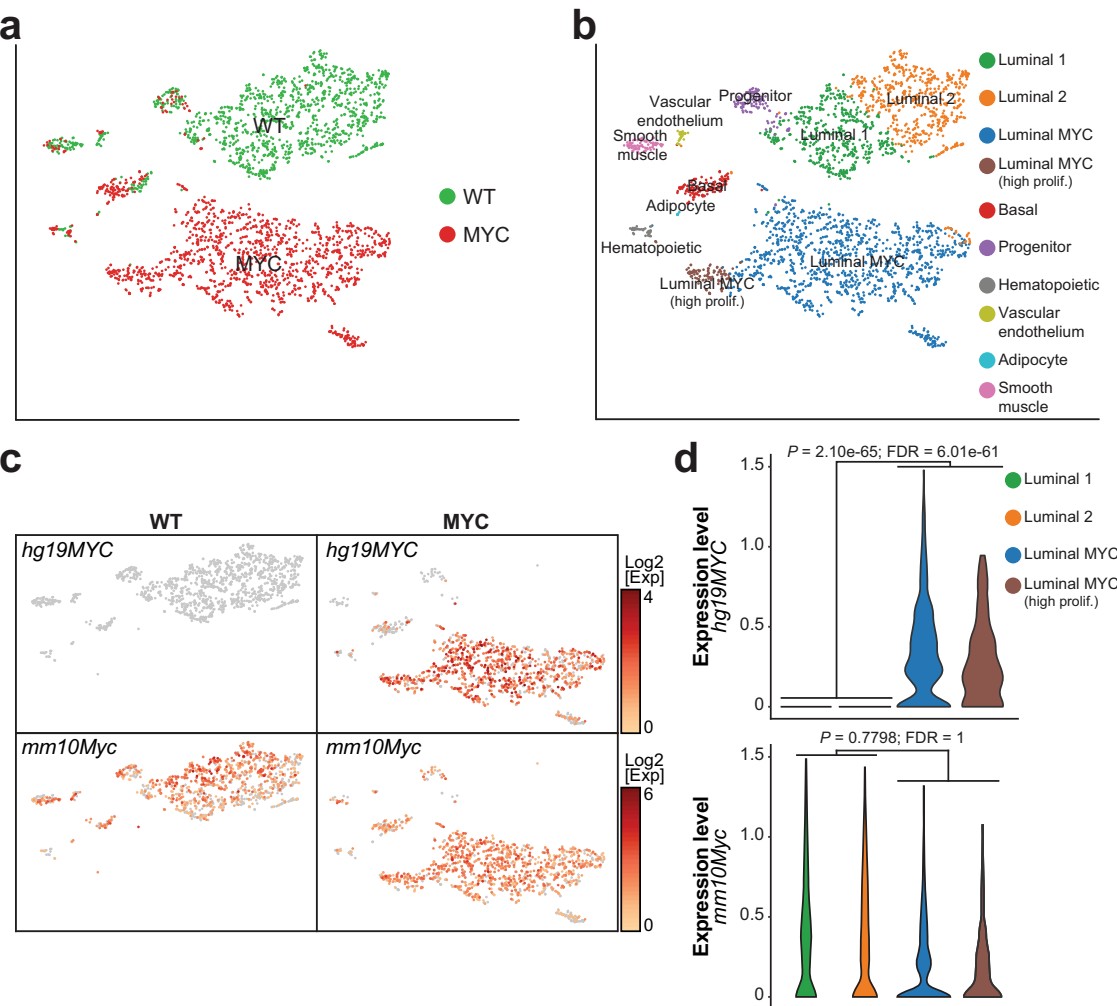

**Fig. 2 Single-cell transcriptome reveals distinct luminal cell subpopulations. a, b** Single cell census of the WT and MYC-transformed VP (**a**) followed by unsupervised clustering revealed four luminal subsets (**b** VP; $n = 1$ per genotype). **c, d** Human *MYC* transcript (*hg19MYC*) is only observed in MYC-transformed VP and mostly restricted to the luminal subsets while murine *Myc* transcript (*mm10Myc*) is expressed across cellular populations and genotypes (**c** VP; $n = 1$ per genotype) and is not correlated with *hg19MYC* expression in luminal cells (**d** VP; $n = 1$ per genotype; edgeR: two-sided quasi-likelihood F-test).

transcriptional program enriched in gene sets related to cell proliferation (E2F_targets, G2M_chekpoint) or MYC-transcriptional activity per se (MYC_targets_V1/V2), was solely driven by the luminal cells (Fig. 3a, b). In fact, the near totality of the MYC-driven transcriptional program captured by bulk RNA-seq is in line with the luminal cells transcriptional program. However, a large proportion of MYC-driven transcriptional reprogramming was undetected in bulk RNA-seq and only captured by single-cell transcriptomics. Notably, basal cells underwent an extensive transcriptional reprogramming (Fig. 3a). Considering that human

*MYC* transgene expression was detected in only a limited proportion of basal cells (18.3%; Fig. 2c), this result suggests the existence of a paracrine transcriptional reprogramming upon MYC overexpression and prostate transformation. In addition, scRNA-seq revealed the downregulation of several transcriptional programs in luminal cells. Critically, the depletion of the Androgen_response gene set (Fig. 3a, c), which was not accompanied with a global decreased in AR transcript and protein levels (Fig. 3d–e; Supplementary Fig. 4e), suggests a dampening of the AR transcriptional program driven by *MYC* overexpression as

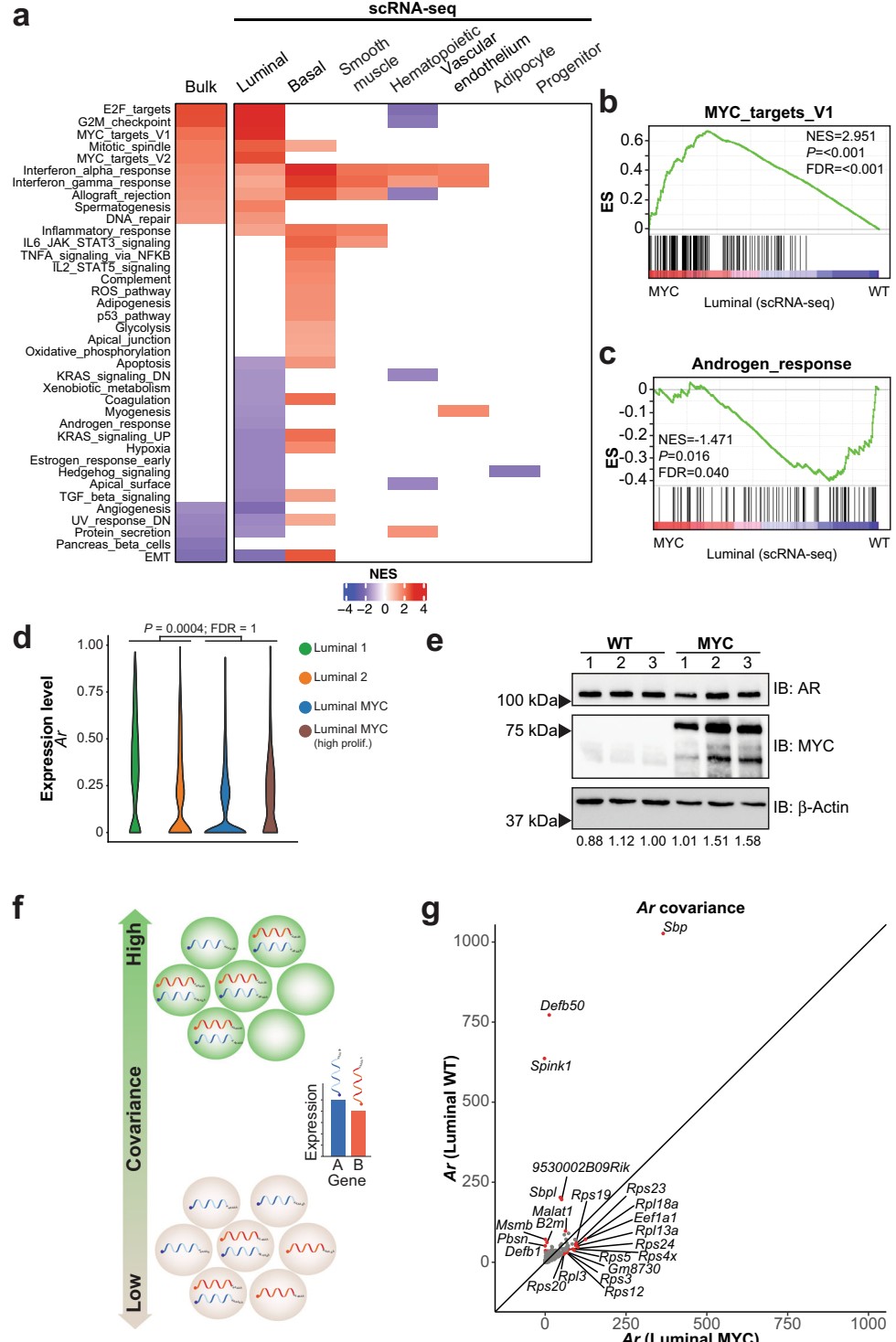

**Fig. 3 MYC-driven luminal cells transformation dampens the AR transcriptional program. a** Gene Set Enrichment Analysis (GSEA, Hallmark, $P < 0.05$ and FDR < 0.1) revealed that the bulk RNA-seq transcriptional program associated with MYC overexpression is mostly driven by the luminal subset (VP; matched bulk and single-cell RNA-seq; $n = 1$ per genotype; Source data are provided as a Source Data file). **b, c** MYC overexpression is associated with an enriched MYC transcriptional program (**b** $P < 0.001$ and FDR < 0.001) and a depleted AR response (**c** $P < 0.016$ and FDR < 0.040) in the luminal subset (GSEA; VP; $n = 1$ per genotype). **d, e** MYC overexpression does not alter AR transcript expression in the luminal compartment (**d** VP; $n = 1$ per genotype; edgeR: two-sided quasi-likelihood F-test) and protein levels in the VP (**e** VP; $n = 3$ per genotype; numbers at the bottom represent AR levels relative to β-Actin; Source data are provided as a Source Data file). **f** Schematic representation of covariance analysis to determine co-expression (i.e. positive covariance) or mutually exclusive expression (i.e. negative covariance) between two genes at a single cell level. **g** Covariance analysis in the luminal subset reveals a shift from canonical AR target genes in the transcripts co-expressed with *Ar* upon MYC overexpression (VP; $n = 1$ per genotype). NES: normalized enrichment score; ES: enrichment score.

exemplified by loss of *Pbsn* and *Msmb* expression in the luminal compartment (Supplementary Fig. 4b)[19,20].

Thus, we sought to leverage single-cell transcriptomics to determine if *MYC* overexpression alters the nature of the transcripts co-expressed with *Ar* through a covariance analysis (Fig. 3f)[23]. As expected, androgen-dependent genes such as *Pbsn*, *Msmb*, *Sbp*, *Defb50* and *B2m* or the prostate-specific *9530002B09Rik* were co-expressed with *Ar* in WT luminal cells (Fig. 3g)[19,20,24–28]. Interestingly, both *Spink1* and *Malat1*, which are respectively associated with castration-resistant or enzalutamide-resistant disease[29,30], were strongly co-expressed with *Ar* only in untransformed tissues (Fig. 3g), suggesting that these genes are also part of the normal androgen-dependent prostate epithelium homeostasis. Surprisingly, upon *MYC* overexpression, canonical AR target genes were no longer co-expressed with *Ar*. Instead, transcripts related to ribosome biogenesis, a key pathway driving cell growth and tumorigenesis and associated with MYC function[31], were co-expressed with *Ar* (Fig. 3g). Altogether, these results indicate that AR-transcriptional program is compromised upon MYC overexpression.

**MYC overexpression alters the AR cistrome**. To further characterize the mechanism whereby MYC overexpression negatively affects the AR-dependent transcriptional program, we utilized chromatin immunoprecipitation followed by high-throughput sequencing (ChIP-seq) to assess the AR cistrome. Although motif analysis of AR binding sites revealed the canonical androgen response element as the top enriched motif across genotypes (Fig. 4a), unsupervised clustering uncovered a distinct AR cistrome driven by MYC overexpression (Fig. 4b). Indeed, MYC overexpression resulted in a significant expansion of the AR cistrome with 1695 sites gained compared to WT tissues (Fig. 4c). Motif analyses revealed that AR gained sites are predominantly associated with the forkhead family of transcription factors motifs (forkhead response elements; FHRE), which includes the established regulator of AR transcriptional activity FOXA1, followed by androgen response elements (ARE; Fig. 4d)[32]. Critically, FOXA1 occupancy was increased at AR gained binding sites in MYC-transformed prostate tissues compared to the WT counterpart ($P = 2.23e{-}62$; Fig. 4e and Supplementary Fig. 6a). Genomic regions gaining AR occupancy were characterized by increased histone H3K27 acetylation (H3K27ac; $P = 4.39e{-}40$; Fig. 4f), a mark of active regulatory regions and transcriptional activity[33], supporting a differential usage of non-coding regulatory elements driven by AR in a MYC overexpressing context. To determine whether the repurposing of the AR cistrome upon MYC overexpression is associated with a distinct transcriptional program, we next integrated AR ChIP-seq to single-cell transcriptomics. Association of 1695 AR binding sites gained upon MYC overexpression (Fig. 4c) to the expression of nearby coding genes in the luminal cell subpopulations, ordered based on slingshot pseudotime inference across genotypes (Supplementary Fig. 6b), highlighted three main expression patterns, namely a MYC-dependent increased, decreased or unchanged expression (Fig. 4g). Using GSEA analysis and the Hallmark gene sets, we identified the MYC_targets_V1 as the top gene set enriched within the set of genes with increased expression. Conversely, we identified the Androgen_response among the gene sets that were significantly enriched within the set of genes with decreased expression (Fig. 4h). Taken together these results indicate, in the context of MYC overexpression, a reprogramming of the AR cistrome that drives an altered transcriptional program.

**Divergent MYC and AR transcriptional programs dictate disease progression**. Since our results in the preclinical model

uncovered a robust interplay between MYC and AR transcriptional programs, we next investigated whether this MYC-driven transcriptional reprogramming is clinically relevant. We used gene expression data to stratify 488 primary prostate cancer patients in the TCGA dataset based on the combined levels of the Hallmark Androgen_response (high; low) and MYC_targets_V1 (high; low) transcriptional signatures[9]. Kaplan-Meier curves revealed that patients bearing a primary tumor characterized by divergent AR and MYC transcriptional programs experienced distinct rates of clinical progression. Tumors characterized by a low AR transcriptional signature with concurrent high MYC transcriptional signature (AR_low/MYC_high) were associated with the shortest time to biochemical recurrence (BCR) while tumors characterized by a high AR transcriptional signature with concurrent low MYC transcriptional signature (AR_high/MYC_low) were associated with the longest time to BCR (Supplementary Fig. 7a, b). Interestingly, concordant AR and MYC transcriptional programs (AR_high/MYC_high; AR_low/MYC_low) were associated with an intermediate time to BCR (Supplementary Fig. 7a, b). Recently, transcriptomic data from nearly 20,000 tumors revealed that patients bearing a localized treatment-naïve primary prostate cancer with low AR-activity (AR-A; based on a signature of nine canonical AR transcriptional targets) experience a shorter time to recurrence[34]. Thus, we next sought to determine if MYC transcriptional activity status in low AR-A tumors could identify a more aggressive subtype of primary prostate cancer using the TCGA dataset. Strikingly, Kaplan-Meier curves revealed that it is the subset of low AR-A tumors with concurrent high MYC transcriptional signature that is associated with a faster time to BCR (AR_low/MYC_high *vs*. AR_low/MYC_low, $P = 0.0001$; Fig. 5a, b). Importantly, we validated this finding in a previously published independent meta-analysis cohort combining 855 patients with individual patient-level data (Fig. 5c and Supplementary Fig. 7c)[35]. Univariable analysis revealed that tumors with AR_low/MYC_high transcriptional signatures are associated with increased rates of BCR (Hazard Ratio (HR) = 1.37, 95% Confidence Interval (CI) 1.03–1.83; $P = 0.030$; Fig. 5d), but this did not remain significant after adjusting for clinicopathologic risk factors in multivariable analysis (Fig. 5d and Supplementary Fig. 7d). Since low AR-A tumors were predicted to be less sensitive to androgen-deprivation therapy and more likely to develop metastatic disease after initial local therapy[34], we next asked whether a high MYC transcriptional activity allows for the identification of a more aggressive subtype of treatment-naïve primary prostate cancer. Strikingly, Kaplan-Meier curves revealed that patients with tumors harboring an AR_low/MYC_high signature were the most likely to develop metastatic disease (Fig. 5e and Supplementary Fig. 7e). Univariable analysis shows that AR_low/MYC_high tumors are associated with an increased risk to develop metastatic disease (HR = 2.93, 95% CI 1.68–5.10; $P < 0.001$; Fig. 5f and Supplementary Fig. 7f). Critically, this finding remained significant in a multivariable competing risks regression analysis adjusting for age, prostate-specific antigen (PSA), Gleason score, surgical margin status, extracapsular extension, seminal vesicles invasion and lymph node involvement (HR = 2.46, 95% CI 1.34–4.52; $P = 0.004$; Fig. 5f and Supplementary Fig. 7f). Altogether, our results suggest that concurrent AR_low/MYC_high transcriptional signatures identify a subgroup of patients that are predisposed to fail standard-of-care therapies and progress to develop metastatic disease.

**High *MYC* expression is associated with a dampened AR transcriptional program and resistance to AR signaling inhibitors in castration-resistant tumors**. CRPC is characterized by

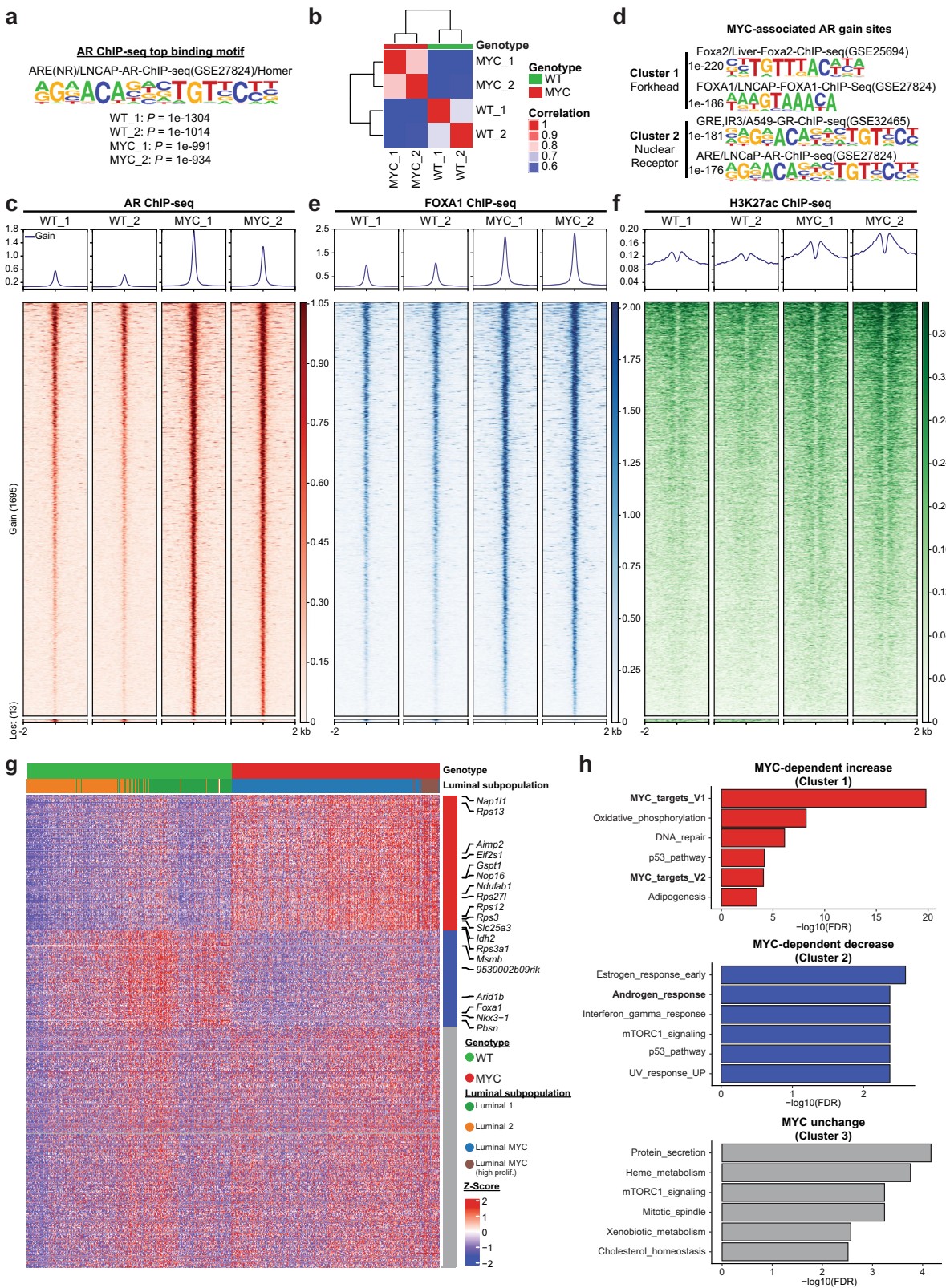

MYC and AR amplification[9,15,36]. Thus, we sought to assess the impact of MYC expression on the AR transcriptional program and cistrome. Gene expression profiling from 59 AR⁺ CRPC tumors revealed that AR activity is negatively correlated with *MYC* expression (Fig. 6a, b)[37]. As expected, GSEA analysis revealed that *MYC*-high CRPC tumors are enriched for MYC transcriptional signatures. Strikingly, the Hallmark

Androgen_response was the only gene set significantly depleted in *MYC*-high tumors (Fig. 6c), supporting a role for MYC in dampening the canonical AR transcriptional program in the castration-resistant setting. We next evaluated whether this phenotype was associated with a repurposing of the AR cistrome using the LuCaP patient-derived xenografts (PDXs) series obtained from AR⁺ mCRPC samples (described in[38] and

**Fig. 4 MYC overexpression alters the AR cistrome. a** AR ChIP-seq identifies an androgen response element (ARE) as the top AR binding motif in WT and MYC-transformed VP (VP; $n = 2$ pools of biological replicates ($n = 8$–13) per genotype). **b** Unsupervised pairwise correlation of the murine AR cistrome from all specimens (VP; $n = 2$ pools of biological replicates ($n = 8$–13) per genotype). **c** MYC overexpression expands the AR cistrome as demonstrated by the heatmaps indicating AR binding intensity across 4 kb intervals (VP; $n = 2$ pools of biological replicates ($n = 8$–13) per genotype). **d** Motif analysis of MYC-associated AR gained sites reveal forkhead response element (FHRE) and androgen response element (ARE; VP; $n = 2$ pools of biological replicates ($n = 8$–13) per genotype). **e, f** AR gained sites are characterized by increased FOXA1 binding (**e**) and H3K27ac mark (**f**) in MYC-transformed VP (VP; $n = 2$ pools of biological replicates ($n = 8$–13) per genotype). **g** Integration of the 1695 AR bindings sites gained in MYC tumors with luminal single cell transcriptome grouped by k-means clustering ($n = 3$ clusters). **h** GSEA analysis (Hallmark) revealed an enforced MYC transcriptional program (Cluster 1) and a diminished androgen response (Cluster 2) associated to MYC-dependent AR gained binding sites (Source data are provided as a Source Data file).

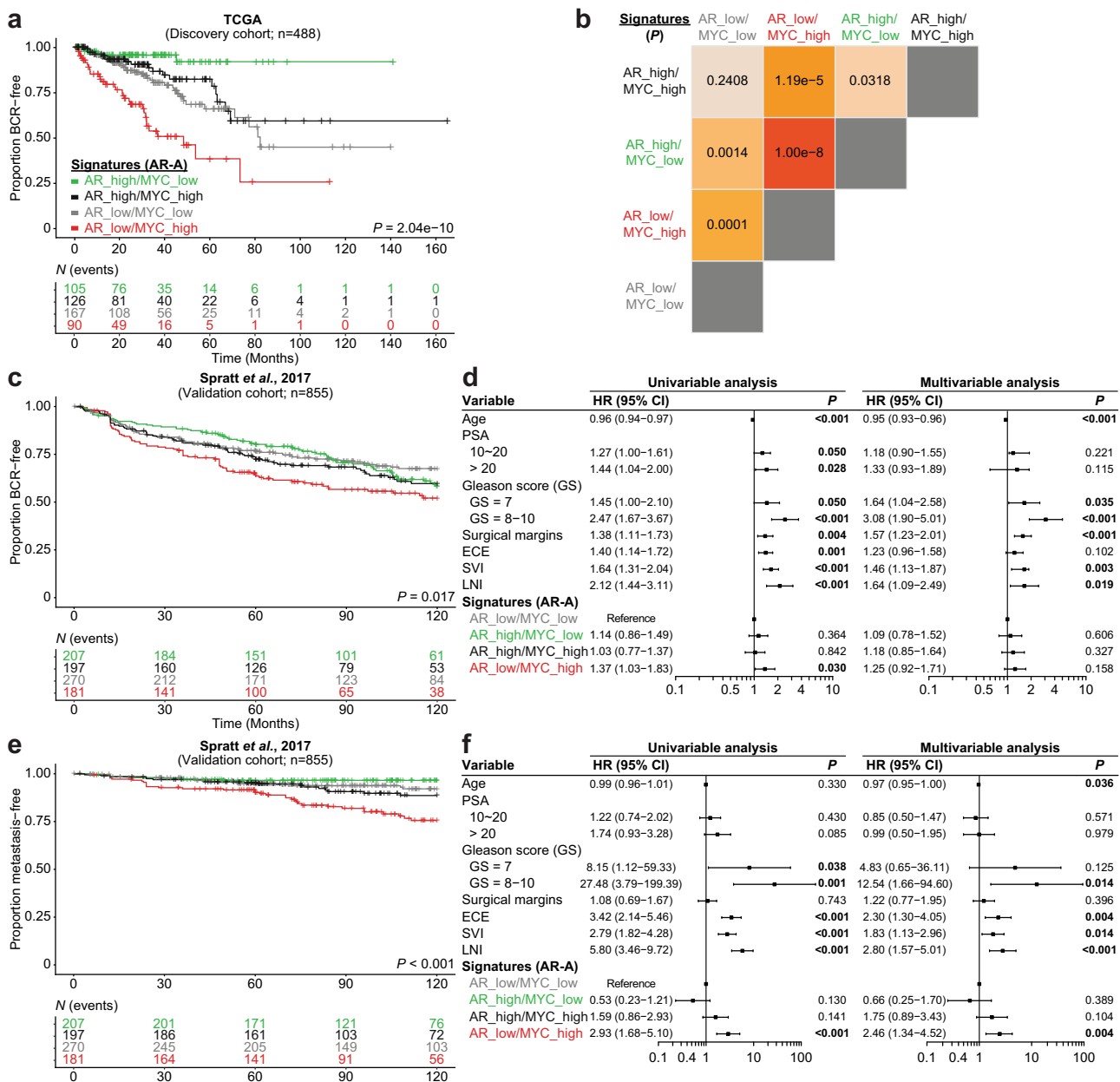

**Fig. 5 Divergent MYC and AR transcriptional programs dictate disease progression. a, b** Kaplan–Meier curves (**a**) and log-rank tests (**b**) reveal that patients bearing a primary tumor characterized by low AR-activity (AR-A) and concurrent high MYC transcriptional signature (Hallmark) have a shorter time to biochemical recurrence (BCR) within the discovery cohort (TCGA). **c, d** Kaplan–Meier curves (**c**), univariable and multivariable analysis (**d** Cox proportional hazards model) confirms that tumors with concurrent low AR-A and high MYC transcriptional signatures develop BCR after radical prostatectomy more rapidly than low AR-A tumors without an active MYC transcriptional program in the validation cohort (Spratt et al., 2017[35]; $n = 855$; HR ± 95% CI). **e, f** Kaplan–Meier curves (**e**), univariable and multivariable analyses (**f** Cox proportional hazards model) reveal that tumors with concurrent low AR-A and high MYC transcriptional signatures are more likely to develop a metastatic disease ($n = 855$; HR ± 95% CI). PSA: prostate-specific antigen; HR: hazard ratio; CI: confidence interval; GS: Gleason score; ECE: extracapsular extension; SVI: seminal vesicles invasion; LNI: lymph node involvement.

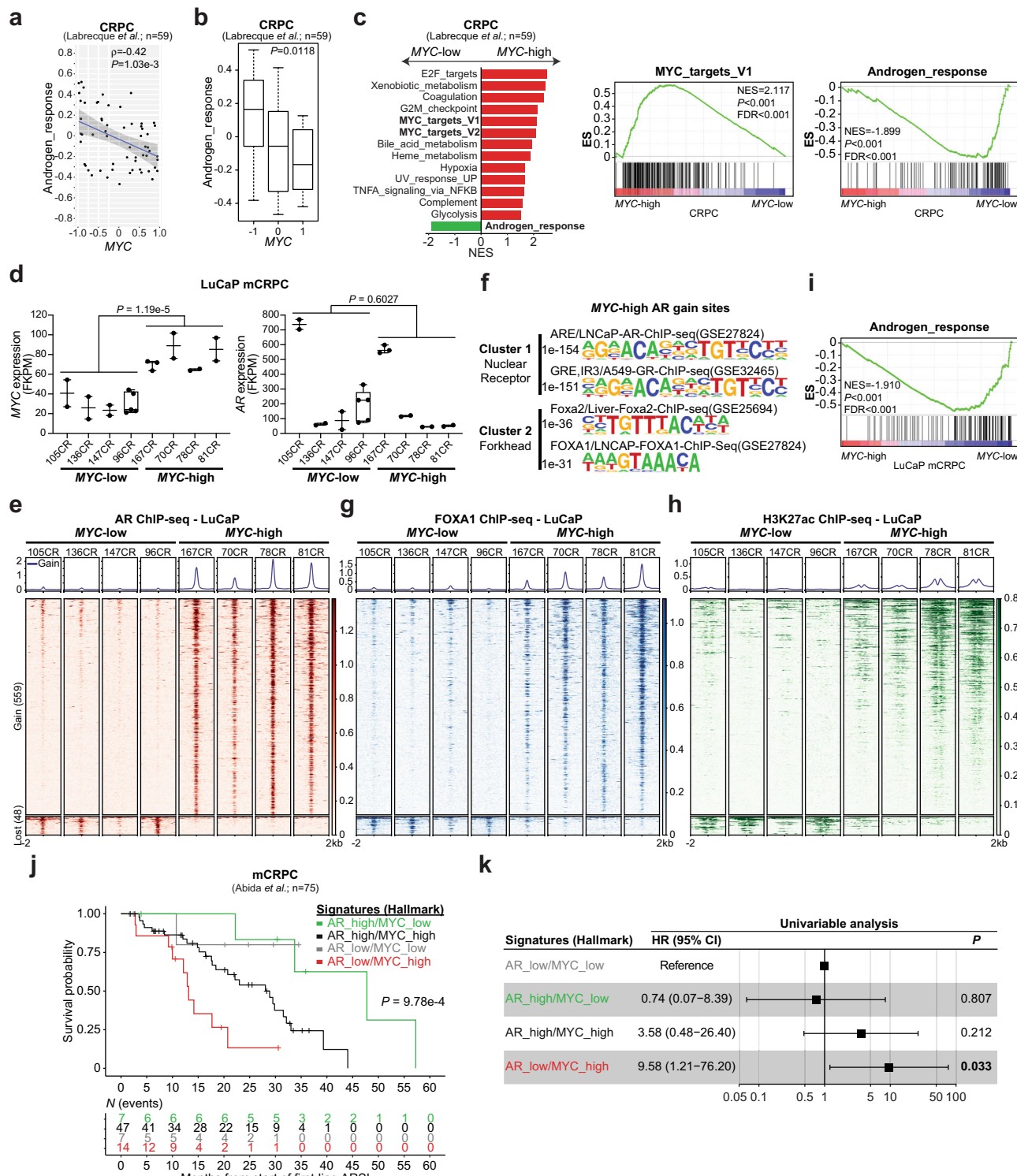

Supplementary Fig. 8a). We selected eight specimens, for which the gene expression profiles were readily available, and stratified them into either the *MYC*-high or the *MYC*-low group based on transcript expression (Fig. 6d)[37]. Importantly, AR transcript level was not different between the *MYC*-high and *MYC*-low groups (Fig. 6d). Comparison of the AR cistrome between the two groups uncovered an alteration of AR binding in *MYC*-high mCRPC PDXs towards an expanded AR cistrome robustly associated with the forkhead family of transcription factors motifs (Fig. 6e, f, Supplementary Fig. 8b). Accordingly, greater FOXA1 occupancy was observed at AR gained binding sites in *MYC*-high compared

to the *MYC*-low mCRPC PDXs (*P* = 1.74e-144; Fig. 6g and Supplementary Fig. 8c). These sites were also characterized by increased H3K27ac mark (*P* = 3.54e-268; Fig. 6h), in agreement with the MYC-driven murine prostate cancer model (Fig. 4). Critically, differential AR chromatin occupancy between both groups was associated with a dampened AR transcriptional program in the *MYC*-high group (Fig. 6i). Considering that high MYC expression dampens the AR transcriptional program, we hypothesized that MYC transcriptional activity is central to the response to next generation ARSI (i.e. abiraterone acetate or enzalutamide) in mCRPC. We used gene expression data to

**Fig. 6 High *MYC* expression is associated with a dampened AR transcriptional program and resistance to AR signaling inhibitors in castration-resistant tumors. a, b** AR activity is inversely correlated with *MYC* expression in CRPC clinical samples (**a** Pearson correlation coefficient (ρ) and *P* value; linear regression ± 95% CI; Source data are provided as a Source Data file) and significantly lower in *MYC*-high tumors (**b** Two-way ANNOVA followed by a Tukey–Kramer test; median; box boundaries: 25th and 75th percentiles; whiskers: ± lowest/smallest value no further than 1.5 interquartile range; *n* = 59; Source data are provided as a Source Data file). **c** Gene Set Enrichment Analysis (GSEA, Hallmark, *P* < 0.05 and FDR < 0.1) revealed an enriched MYC transcriptional program (*P* < 0.001 and FDR < 0.001) and a depleted AR response (*P* < 0.001 and FDR < 0.001) in *MYC*-high CRPC (Source data are provided as a Source Data file). **d, e** *MYC*-high mCRPC LuCaP patient-derived xenografts (PDXs) have similar levels of *AR* (**d** Wilcoxon rank-sum test; *n* = 8 biologically independent animals; median, whiskers ± min to max; Source data are provided as a Source Data file) but are associated with an expanded AR cistrome as demonstrated by the increased binding intensity across 4 kb intervals at AR gained sites (**e**). **f** Motif analysis of *MYC*-associated AR gained sites reveal ARE and FHRE. **g, h** AR gained sites are characterized by increased FOXA1 binding (**g**) and H3K27ac mark (**h**) in *MYC*-high mCRPC LuCaP. **i** AR cistrome in *MYC*-high mCRPC LuCaP PDXs is associated with a diminished androgen response (GSEA; *P* < 0.001 and FDR < 0.001). **j, k** Kaplan-Meier curves (**j**) and univariable analysis (**k** Cox proportional hazards model) revealed that patients with mCRPC tumors harboring an AR_low/MYC_high signature are more likely to resist ARSI treatment and die of their disease (*n* = 75; HR ± 95% CI). HR: hazard ratio; CI: confidence interval; NES: normalized enrichment score; ES: enrichment score.

stratify 75 mCRPC in the SU2C International Dream Team dataset based on the combined levels of the Hallmark Androgen_response (high; low) and MYC_targets_V1 (high; low) transcriptional signatures[39]. Strikingly, Kaplan–Meier curves and univariable analysis revealed that patients with mCRPC tumors harboring an AR_low/MYC_high signature were more likely to resist ARSI treatment and die of their disease (HR = 9.58, 95% CI 1.21–76.20; *P* = 0.033 Fig. 6j, k). Taken together, these results support the existence of a distinct AR cistrome in *MYC* over-expressing CRPC associated with a diminished AR transcriptional program and suggest that concurrent AR_low/MYC_high transcriptional signatures identify a subgroup of patients that are predisposed to fail first-line next generation ARSI treatment and die of mCRPC.

**MYC overexpression disrupts the AR transcriptional program by pausing AR regulated genes.** To assess for direct effects of AR in mediating this transcriptional reprogramming we leveraged the preclinical model of MYC-driven prostate cancer and performed binding and expression target analysis (BETA) to integrate MYC-driven gene expression changes in murine VP with genome-wide AR binding data[40]. This analysis revealed that AR binding was significantly associated with genes downregulated by MYC overexpression (*P* = 2.32e-5; Fig. 7a). Along this line, AR binding was found to be increased at genomic regions nearby Andro-gen_response genes alongside the H3K27ac mark following MYC overexpression (Fig. 7b, c), in contrast with the accompanied depletion of the Androgen_response gene set (Fig. 3c). For example, AR and FOXA1 binding was increased in the promoter region of *Pbsn* (Fig. 7d), an AR-dependent gene whose transcript level was severely downregulated following MYC overexpression (Fig. 7e; Supplementary Fig. 4b). In the promoter region of *Msmb*, another AR-dependent gene previously characterized as a tumor suppressor[41], AR and FOXA1 binding as well as the H3K27ac mark levels were maintained although *Msmb* transcript levels were also downregulated by MYC overexpression (Fig. 7f, g and Supplementary Fig. 4b). These results suggest that MYC-driven repression of the AR transcriptional program is not associated with a disengagement of AR or the loss of the H3K27ac mark.

Using the androgen responsive LNCaP prostate cancer cell line, Barfeld and colleagues have previously reported that MYC overexpression antagonizes the transcriptional activity of the AR[15]. Similarly to the MYC-driven genetically engineered prostate cancer mouse model, MYC overexpression in LNCaP cells was associated with the depletion of the Hallmark Androgen_response gene set (Supplementary Fig. 9a). Annotation of the AR cistrome and gene expression data by BETA revealed that AR binding is associated with downregulated genes, supporting a global reduction in AR transcriptional activity

driven by MYC overexpression. Conversely, MYC cistrome was predominantly associated with upregulated genes, consistent with its role as a transcriptional activator (Supplementary Fig. 9b). Again, AR binding nearby Androgen_response genes remained largely unchanged following MYC overexpression. Interestingly, MYC binding nearby MYC_targets_V1 genes also remained unchanged following MYC overexpression despite a significant enrichment of the MYC_targets_V1 gene set (Supplementary Fig. 9c). Inspection of AR and MYC binding in the vicinity of canonical AR-dependent genes such as *KLK3* and *TMPRSS2* also revealed unchanged binding profiles (Supplementary Fig. 9d).

Based on the evidence for MYC regulation of RNA Pol II pause release[42], we leveraged RNA Pol II ChIP-seq to determine genome-wide RNA Pol II traveling ratio (*i.e.* RNA Pol II density in the promoter-proximal region over the RNA Pol II density in the transcribed region) in vivo following MYC overexpression in murine VP (Fig. 7h). As expected, genes with reduced RNA Pol II traveling ratio following MYC overexpression were enriched for MYC transcriptional signatures, indicative of pause release at these sites (Fig. 7i, j and Supplementary Fig. 10a). Critically, genes with greater RNA Pol II traveling ratio were enriched for the AR transcriptional signature, suggestive of enhanced RNA Pol II pausing at AR-regulated genes (Fig. 7k, l and Supplementary Fig. 10a). Along this line, ChIP-seq revealed a build-up of RNA Pol II occupancy at the promoter of the AR-regulated gene *Pbsn* following MYC overexpression (Fig. 7m). At the *Msmb* locus, another AR-regulated gene, RNA Pol II occupancy remained unchanged at the promoter region but was abrogated at the gene body in the MYC overexpressing condition (Fig. 7n). These features are in stark contrast to MYC-regulated genes such as *Rps3* and *Rps5* for which we observed an increase RNA Pol II occupancy at the gene body in the MYC overexpressing condition (Supplementary Fig. 10b, c). Since these patterns suggest a MYC-driven altered ratio of initiating and elongating RNA Pol II at AR-regulated genes, we next determined the RNA Pol II traveling ratio at Androgen_response genes. Strikingly, RNA Pol II traveling ratio at Androgen_response genes was significantly increased by MYC overexpression (*P* = 0.0021; Fig. 8a and Supplementary Fig. 10d), supporting MYC-driven RNA Pol II promoter-proximal pausing and consequently non-productive transcription at AR-dependent genes. Altogether these findings support RNA Pol II promoter-proximal pausing as a potential mechanism for MYC-mediated transcriptional repression at AR regulated genes associated with the canonical AR transcriptional signature (Fig. 8b)[43].

**Discussion**

In this study, we report the impact of MYC overexpression in vivo on the AR transcriptional program. By leveraging the expression of a human MYC transgene (*hg19MYC*) observed at a

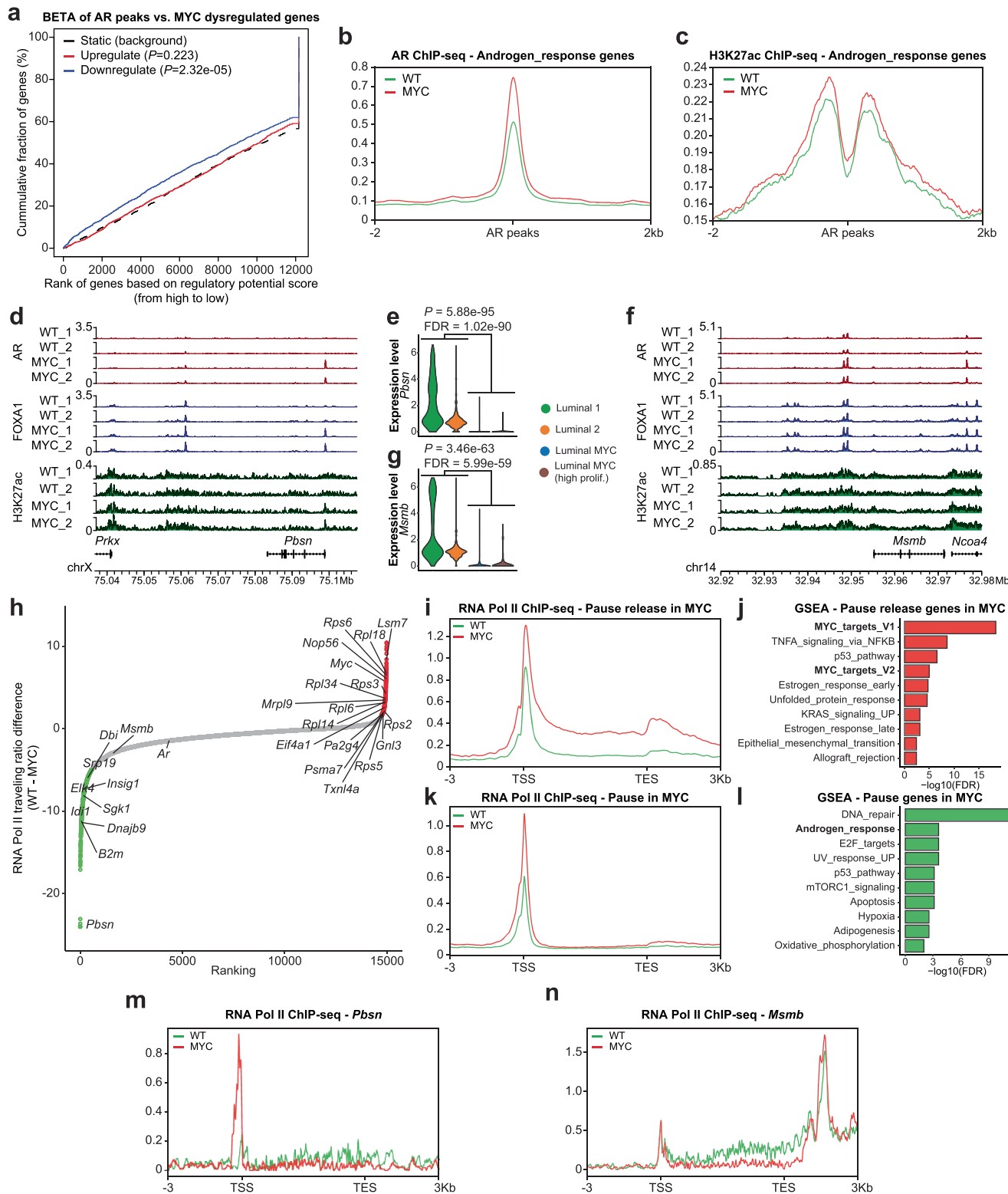

single-cell level in murine prostatic tissues, our data demonstrate that MYC overexpression robustly reprograms luminal (Krt8[Hi], Krt18[Hi]) cells toward a repressed AR transcriptional program, a feature contrasting with the supporting role of MYC on the AR transcriptional program in the apocrine breast cancer subtype[44]. Our single-cell transcriptome data delineate a minor luminal subpopulation expressing high levels of Cd44, Tacstd2 (Trop2) and Psca markers associated with luminal progenitor cells[18]. Recently, single-cell transcriptomics performed in the murine AP lobe also

revealed a distinct but rare luminal subpopulation anatomically lining the proximal duct and expressing Tacstd2 (Trop2), Psca as well as Ly6a (Sca-1), Krt4 and Cldn10[45]. An independent study suggested that the luminal subpopulation expressing high levels of progenitor markers such as Tacstd2 (Trop2), Psca, Ly6a (Sca-1) and Krt4 corresponds to urethral luminal cells extending into the proximal ducts of the prostate[46]. Since the luminal progenitor population identified in the VP lobe expressed all the aforementioned markers (Supplementary Fig. 4a, f), we cannot rule out the

**Fig. 7 MYC overexpression disrupts the AR transcriptional program by pausing AR regulated genes. a** BETA analysis revealed that AR binding sites are associated with gene downregulation following MYC overexpression. **b, c** Despite a dampened AR transcriptional program, higher levels of the AR binding (**b**) and H3K27ac mark (**c**) are observed nearby AR response genes (VP; $n = 2$ pools of biological replicates ($n = 8$–13) per genotype). **d, e** AR, FOXA1 and H3K27ac tracks at *Pbsn* locus, an AR-dependent gene, reveal unchanged or heightened AR and FOXA1 binding (**d** VP; $n = 2$ pools of biological replicates ($n = 8$–13) per genotype) albeit decreased transcript level (**e** VP; $n = 1$ per genotype; edgeR: two-sided quasi-likelihood F-test) following MYC overexpression. **f, g** Unchanged AR and FOXA1 binding and H3K27ac mark at *Mmsb* locus (**f** VP; $n = 2$ pools of biological replicates ($n = 8$–13) per genotype), an AR-dependent gene downregulated by MYC overexpression (**g** VP; $n = 1$ per genotype; edgeR: two-sided quasi-likelihood F-test). **h** RNA Pol II traveling ratio differences following MYC overexpression in murine VP (VP; $n = 2$ pools of biological replicates ($n = 8$–13) per genotype). **i, j** Pause release genes following MYC overexpression are characterized by greater RNA Pol II occupancy at gene body (**i** VP; $n = 2$ pools of biological replicates ($n = 8$–13) per genotype) and are enriched for MYC transcriptional signatures (**j** GSEA, Hallmark, $P < 0.05$ and FDR < 0.1; Source data are provided as a Source Data file). **k, l** Pause genes following MYC overexpression are characterized by greater promoter-proximal RNA Pol II occupancy (**k** VP; $n = 2$ pools of biological replicates ($n = 8$–13) per genotype) and are enriched for AR transcriptional signature (**l** GSEA, Hallmark, $P < 0.05$ and FDR < 0.1; Source data are provided as a Source Data file). **m, n** Increased RNA Pol II occupancy at the promoter of *Pbsn* (**m**) and decreased occupancy at the gene body of *Msmb* (**n**) following MYC overexpression (VP; $n = 2$ pools of biological replicates ($n = 8$–13) per genotype). TSS: transcription start site; TES: transcription end site.

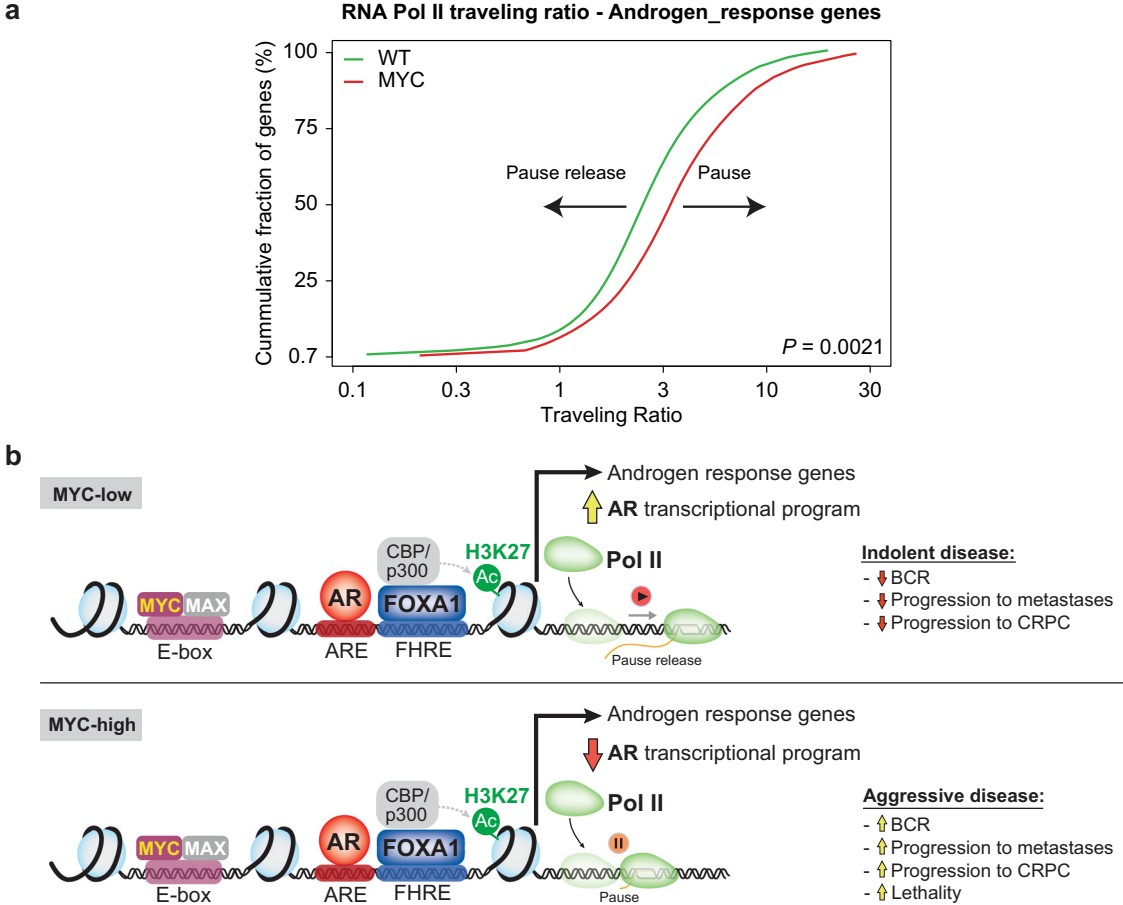

**Fig. 8 MYC disrupts transcriptional pause release at androgen receptor targets. a** RNA Pol II traveling ratio reveals greater promoter-proximal pausing at Androgen_response genes (two-tailed *t*-test). **b** Graphical summary. TSS: transcription start site; TES: transcription end site; BCR: biochemical recurrence; CRPC: castration resistant prostate cancer; FHRE: forkhead response elements; ARE: androgen response elements.

possibility that they might be of urethral origin. Regardless, these progenitor cells were not transcriptionally reprogramed following MYC overexpression (Fig. 3a).

In analyzing the expression of *hg19MYC* transcript driven by the ARR$_2$Pb promoter we found it was not detected in WT prostates, as expected. Surprisingly, we detected low, but consistent *hg19MYC* expression in non-luminal subpopulations (basal: 17/93 (18.3%); hematopoietic: 3/35 (8.6%); vascular endothelium: 1/8 (12.5%); Fig. 2c). While the ARR$_2$Pb promoter used to drive *hg19MYC* expression has been described as highly specific for prostatic epithelium[14,20,47], our single-cell

transcriptome highlights a potentially underappreciated leaky expression of ARR$_2$Pb-driven transgene. However, these seemingly stochastic events are likely transient since Hi-MYC mice do not develop other MYC-driven malignancies, such as B-cell leukemia/lymphoma[48]. With the increasing availability of single-cell transcriptomic profiles from various genetically engineered mouse models (GEMMs), it is expected that tissue specific promoter specificity will be reassessed through a new lens.

MYC is commonly amplified in primary prostate cancer and is overexpressed in 37% of metastatic disease[9,49]. Considering that prostate cancer cells that develop resistance to AR-targeted

therapy usually maintain AR expression[50,51], the interplay between MYC and AR is likely to remain critical as the disease progress to the CRPC stage. Importantly, our analyses exposed a subtype of primary prostate cancer characterized by divergent AR (low) and MYC (high) transcriptional signatures that are predisposed to fail standard-of-care therapies and progress to the mCRPC stage (Fig. 5). Arriaga and colleagues have recently reported a MYC and RAS co-activation signature associated with metastatic progression and failure to anti-androgen treatments[52]. It is thus tempting to speculate that MYC decreases the reliance of prostate cancer cells on the canonical AR transcriptional program, therefore facilitating resistance to AR-targeted therapies. Along this line, we found that patients harboring a mCRPC characterized by divergent AR (low) and MYC (high) transcriptional signatures are more likely to fail first-line next generation ARSI treatment (i.e. abiraterone acetate or enzalutamide) and die of their disease. In support of c-MYC mediating resistance to ARSI treatment, Bai et al. recently showed that a c-Myc inhibitor disrupting c-Myc and Max dimerization sensitizes enzalutamide-resistant prostate cancer cells to growth inhibition by enzalutamide[53]. Considering that transition from CRPC to neuroendocrine prostate cancer (NEPC) is driven by N-Myc, which also abrogates AR transcriptional program, and that N-Myc is functionally complementary to c-Myc in various processes[54,55], it is now evident that Myc family members are key to prostate cancer etiology and resistance to standard-of-care therapies. These results support the use of therapies not centered on the inhibition of AR signaling (e.g. PARP inhibitors, [177Lu]Lu-PSMA-617) for the subgroup of patients harboring concurrent AR_low/MYC_high transcriptional programs.

Intriguingly, although MYC overexpression antagonizes the AR transcriptional program, this was not associated with a diminished but rather an expanded AR cistrome, characterized by FOXA1 co-occupancy and an active chromatin state. Data from our MYC-driven prostate cancer mouse model, together with a previously published LNCaP model engineered to overexpress MYC, revealed that MYC-driven repression of the AR transcriptional program is not associated with a disengagement of AR or the loss of the H3K27ac mark. Rather, we observed greater RNA Pol II promoter-proximal pausing and non-productive transcription at AR-dependent genes repressed by MYC in vivo. Importantly, no evidence of direct interaction between MYC and AR has been found[15,53], suggesting that the suppression of the AR transcriptional program is not guided by a physical interaction with MYC but rather by a MYC-induced RNA Pol II pausing overcoming the AR enhancers driving AR-regulated genes. Taken together, these results support cofactor redistribution driven by increased MYC expression and resulting in greater RNA Pol II promoter-proximal pausing as a potential mechanism for MYC-mediated transcriptional repression at genes regulated specifically by the AR (Fig. 8b)[43,56].

Altogether, our study revealed an intricate crosstalk between the AR, MYC, FOXA1 and RNA Pol II resulting in a corrupted AR transcriptional program and promoting prostate cancer initiation and progression to the mCRPC stage. Considering that a simple dietary intervention meant to reduce saturated fat consumption can dampen MYC transcriptional program, and the recent development of viable MYC inhibitors for therapeutic interventions[17,57], we foresee that targeting MYC may help restore a canonical AR transcriptional program and sensitize prostate cancer to AR-targeted therapies.

## Methods

**Animal husbandry**. FVB Hi-MYC mice (strain number 01XK8), expressing the human c-MYC transgene in prostatic epithelium, were obtained from the National Cancer Institute Mouse Repository at Frederick National Laboratory for Cancer Research[14]. Upon weaning (3 weeks), male mice heterozygous for the transgene (MYC), together with their wild type littermates (WT), were fed a purified diet (TD.130838, Envigo). Animals were kept on a 12-hour light / 12-hour dark cycle, and allowed free access to food and water at the Dana-Farber Cancer Institute (DFCI) Animal Resources Facility (housing ambient temperature: 22 °C ± 2 °C; ambient humidity: 30–70%). The animal protocol was reviewed and approved by the DFCI Institutional Care and Use Committee (IACUC), and was in accordance with the Animal Welfare Act. For protein expression experiments, mice were housed in the Animal Resources Facility at the Research Institute of the McGill University Health Centre (RI-MUHC) where they were fed a regular lab chow (T.2918, Envigo) from the time of weaning (housing ambient temperature: 21 °C ± 1 °C; ambient humidity: 40–60% ± 5%). The animal protocol followed the ethical guidelines of the Canadian Council on Animal Care, and was approved by the RI-MUHC Glen Facility Animal Care Committee (FACC). Tumor burden in male MYC mice is not associated with adverse effects before the experimental end point (i.e. 12 weeks of age)[14].

*Genotyping*. Tail snips were sent to Transnetyx (Transnetyx, Inc.) for genotyping or genomic DNA was extracted from ear punches using 0.4 mL of lysis buffer (100 mM Tris-HCl pH 7.5, EDTA 5 mM, 2% SDS, 200 mM NaCl and 100 μg/μL freshly added Proteinase K). Samples were incubated overnight at 52 °C. After centrifugation at 10,000 × g for 20 min, the supernatant was collected and mixed by inversion with 0.4 mL isopropanol to precipitate the DNA, which was pelleted by centrifugation for 5 min, then washed with 0.5 mL 70% ethanol and dissolved in 10 μL molecular grade water. The presence of the MYC transgene was detected by polymerase chain reaction (PCR), using the following primer combination: primer 1: 5' AAA CAT GAT GAC TAC CAA GCT TGG C 3' and primer 2: 5' ATG ATA GCA TCT TGT TCT TAG TCT TTT TCT TAA TAG GG 3'. PCR products were resolved using a 2% agarose tris-acetate-EDTA gel and a 177 bp band was visualized using the ChemiDoc™ imaging system (Bio-Rad).

### Tissue specimens

*FVB Hi-MYC model*. At 12 weeks of age, male mice were euthanized by $CO_2$ / isoflurane followed by cervical dislocation. Mouse prostate lobes (AP, DLP, VP) were dissected, weighed and immediately processed for bulk and single-cell transcriptomics or flash-frozen in liquid nitrogen for chromatin immunoprecipitation or protein expression experiments. Tissues were consistently collected during the same periods to minimize inter-samples and circadian rhythm variability.

*mCRPC LuCaP PDXs*. Informed consent was obtained to collect human mCRPC tissues and generate the patient-derived xenograft tumors as described previously (male CB17 SCID mice between 4–6 weeks of age; maximum tumor size: 1000 mm³; housing ambient temperature: 20–26 °C; ambient humidity: 30–70%)[37,38]. The study was approved by the University of Washington Human Subjects Division institutional review board (no. 2341). All animal studies were approved by University of Washington IACUC and performed according to NIH guidelines. Molecular characterization of AR+ mCRPC LuCaP PDXs 70CR, 78CR, 81CR, 96CR, 105CR, 136CR and 147CR was previously described[37,38]. LuCaP PDX 167CR was established from a liver metastasis of a male who died of abiraterone-, carboplatin- and docetaxel-resistant CRPC. LuCaP 167CR expresses AR (mouse monoclonal [F39.4.1] anti-AR; #MU256-UC, Biogenex; dilution 1:60), responds to castration and is negative for synaptophysin (mouse monoclonal [D-4] anti-synaptophysin; #sc-17750, Santa Cruz Biotechnology; dilution 1:200). PDX cellular morphology recapitulates the original liver metastasis (Supplementary Fig. 8a; characterization as previously described[37]).

### Bulk RNA-sequencing

*FVB Hi-MYC model*. Fresh prostate lobes from 12-week-old mice were dissociated to form a single cell suspension. Prostate lobes were minced with a sterile razor blade and resuspended in collagenase/hyaluronidase (#07912, Stemcell Technologies) diluted in DMEM/F-12 (#36254, Stemcell Technologies) at 37 °C for 2 h. After dissociation, cells were centrifuged (350 × g for 5 min) and resuspended in 5 mL of prewarmed 0.25% trypsin/EDTA (#07901, Stemcell Technologies) at 37 °C for 5 min. Trypsinization was stopped with 10 mL of cold HBSS (#37150, Stemcell) supplemented with 2% of regular cell culture grade FBS. Cells were centrifuged (350 × g for 5 min) and resuspended in 1 mL of prewarmed dispase (#07913, Stemcell Technologies) and 100 μL of DNase I (#07900, Stemcell Technologies) and passed 5 times through a 27 G syringe needle. Cells were then mixed with 10 mL of cold HBSS supplemented with 2% FBS, filtered through a 40 μm cell strainer (#27305, Stemcell Technologies), centrifuged (350 × g for 10 min) and resuspended in PBS. An aliquot of the single cell suspension was immediately processed for single-cell RNA-sequencing and RNA from an equal number of cells was extracted using the miRNeasy Micro Kit (#217084, Qiagen) coupled with on-column DNAse treatment (#79254, Qiagen) for bulk RNA-sequencing. RNA sample concentration was measured and subjected to quality evaluation, using a Bioanalyzer RNA 6000 Nano kit (#5067-1511, Agilent). The Dana-Farber Cancer Institute Molecular Biology Core Facilities prepared libraries from 500 ng of purified total RNA, using TruSeq Stranded mRNA sample preparation kits (#RS-122-2101, Illumina) according to the manufacturer's protocol. Finished libraries were quantified by the

Qubit dsDNA High-Sensitivity Assay Kit (#32854, Thermo Fisher Scientific), by an Agilent TapeStation 2200 system using D1000 ScreenTape (#5067-5582, Agilent), and by RT-qPCR using the KAPA library quantification kit (#KK4835, Kapa Biosystems), according to the manufacturers' protocols; pooled uniquely indexed RNA-seq libraries in equimolar ratios were sequenced to a target depth of 40 M reads on an Illumina NextSeq500 run with single-end 75 bp reads. Read alignment, quality control and data analysis was performed using VIPER (2.0)[58], RNA-seq reads were mapped by STAR (2.7.0f)[59] and read counts for each gene were generated by Cufflinks (2.2.1)[60]. Differential gene expression analyses were performed on absolute gene counts for RNA-seq data and raw read counts for transcriptomic profiling data using DESeq2 (1.18.1)[61].

*mCRPC LuCaP PDXs.* LuCaP PDX tumor samples were collected from castrated CB 17 SCID male mice. Frozen tumors were used for RNA extraction and RNA-seq analysis as described previously[37].

*LNCaP MYC model.* Published gene expression data (GSE73995[15]) was downloaded and reanalyzed.

**Single-cell RNA-sequencing.** Cell preparation for 3' barcoded scRNA-seq (#120237, Chromium V2 assay) was performed according to the manufacturer's protocol (10X Genomics) targeting 5000 cells from single-cell suspensions of freshly processed prostate lobes as described above. Single-cell RNA-seq data were preprocessed using the 10x genomics Cell Ranger (https://www.10xgenomics.com; 2.0.0) to obtain the UMI (unique molecular identifier) counts for each gene. To get a reliable single cell transcriptome dataset, we excluded the cells with fewer than 200 genes expressed (UMI > 0) or the cells with more than 80% UMIs from mitochondrial genes. The filtered data was then normalized and scaled by using seurat R package (3.1.1) to remove unwanted sources of variations[62]. tSNE was performed on the normalized data to visualize the single cells in two-dimensional space by using the result of principal component analysis (PCA). Unsupervised clustering was performed by using the "FindClusters" function in the seurat R package (3.1.1) with parameters of resolution = 0.8. Genes with differential expression between clusters were obtained by using Wilcoxon rank-sum test. FDR was calculated to correct for multiple testing.

*Specific gene expression levels.* The normalized expression level for all cells was calculated by the seurat R package (3.1.1). The Violin plots were created by the geom_violin function in ggplot2 R package (3.3.2), scale option set to 'area'.

*Covariance analysis.* The covariance for all genes with *Ar* is calculated by the cov function in stats R package (3.6.0). Genes that have covariance difference larger than 30 between the WT and MYC samples were colored in red and labeled in the plot.

*Slingshot pseudotime inference.* Pseudotime inference is done by the slingshot R package (1.3.1). K-means clustering results and tSNE coordinates were used as input for the pseudotime inference.

**Bioinformatics analyses – bulk RNA-seq and scRNA-seq**
*Bulk RNA-seq and scRNA-seq gene expression correlation.* X-axis is the log(scRNA-seq sum of UMI from all cells), Y-axis is log(bulk RNA-seq − raw read counts). Correlation is calculated based on Pearson correlation. The Venn diagram is the overlap expressed genes between scRNA-seq and bulk RNA-seq. A gene is considered as expressed when the sum of UMI from all cells is larger than 0 in scRNA-seq or raw read counts is larger than 0 in bulk-RNA-seq.

*Sample-sample correlation and principal component analysis (PCA).* Sum of UMI from all cells in scRNA-seq and raw read counts in bulk RNA-seq for matched samples were calculated. Batch effects between scRNA-seq and bulk RNA-seq data were removed using the ComBat approach from SVA (3.18.00). Pearson correlation and principal components were calculated using the counts after removal of batch effect.

*Gene set enrichment analysis (GSEA).* All GSEA were done using pre-ranked analysis (GSEA Java; v4.1.0) with Hallmark gene sets (h.all.v7.2.symbols.gmt). Heatmap visualization of normalized enrichment score (NES) was obtained using ComplexHeatmap R package (2.2.0)[63].

**Protein expression.** Fresh-frozen VP tissues from 12-week-old male FVB mice were sliced on ice with stainless steel disposable scalpels (Fisher Scientific) then homogenized in RIPA buffer (20 mM Tris-HCl pH 7.5, 150 mM NaCl, 1 mM EDTA, 1% TRITON-X) supplemented with phosphatases and protease inhibitors (Mini, Pierce™, Thermo Fisher) using a tissue grinder kit (Kontes). Equal amounts of protein (15 µg; Pierce™ Rapid Gold BCA Protein Assay, Thermo Fisher) were resolved on 8–12% Tris-glycine SDS-polyacrylamide gels and transferred to nitrocellulose blotting membranes (Bio-Rad), following standard procedures.

Membranes were probed with the following antibodies according to the manufacturer's instructions: rabbit monoclonal [Y69] anti-c-MYC (#ab32072, Abcam; dilution 1:1,000), rabbit monoclonal [ER179(2)] anti-AR (#ab108341, Abcam; dilution 1:1,000) or rabbit polyclonal anti-β-Actin (#4967, Cell Signaling Technology; dilution 1:1,000). Densitometry analyses were made with ImageJ (U.S. NIH, Bethesda, MD; http://imagej.nih.gov/ij/). Results were normalized to β-actin and expressed as arbitrary units.

**ChIP-sequencing**
*FVB Hi-MYC model.* ChIP-sequencing was performed as described in Labbé and Zadra et al.[17]. Briefly, fresh-frozen VP tissues from 12-week-old mice were pulverized (Cryoprep Impactor, Covaris), resuspended in PBS + 1% formaldehyde, and incubated at room temperature for 20 min. Fixation was stopped by the addition of 0.125 M glycine (final concentration) for 15 min at room temperature, then washed with ice cold PBS + EDTA-free protease inhibitor cocktail (PIC; #04693132001, Roche). Multiple biological replicates were combined for each condition in two distinct pools (replicates). Chromatin was isolated by the addition of lysis buffer (0.1% SDS, 1% Triton X-100, 10 mM Tris-HCl (pH 7.4), 1 mM EDTA (pH 8.0), 0.1% NaDOC, 0.13 M NaCl, 1X PIC) + sonication buffer (0.25% sarkosyl, 1 mM DTT) to the samples, which were maintained on ice for 30 min. Lysates were sonicated (E210 Focused-ultrasonicator, Covaris) and the DNA was sheared to an average length of ~200–500 bp. Genomic DNA (input) was isolated by treating sheared chromatin samples with RNase (30 min at 37 °C), proteinase K (30 min at 55 °C), de-crosslinking buffer (1% SDS, 100 mM NaHCO3 (final concentration), 6–16 h at 65 °C), followed by purification (#28008, Qiagen). DNA was quantified on a NanoDrop spectrophotometer, using the Quant-iT High-Sensitivity dsDNA Assay Kit (#Q33120, Thermo Fisher Scientific). On ice, AR (2 µg, #ab108341, Abcam), FOXA1 (6 µg, #ab23738, Abcam), RNA Pol II (4 µg, #sc899, Santa Cruz Biotechnology) or H3K27ac (10 µl, #ab4729, Abcam) antibodies were conjugated to a mix of washed Dynabeads protein A and G (Thermo Fisher Scientific), and incubated on a rotator (overnight at 4 °C) with 5 µg (AR, FOXA1, RNA Pol II) or 1.5 µg (H3K27ac) of chromatin. ChIP'ed complexes were washed, sequentially treated with RNase (30 min at 37 °C), proteinase K (30 min at 55 °C), de-crosslinking buffer (1% SDS, 100 mM NaHCO3 (final concentration), 6–16 h at 65 °C), and purified (#28008, Qiagen). The concentration and size distribution of the immunoprecipitated DNA was measured using the Bioanalyzer High Sensitivity DNA kit (#5067-4626, Agilent). Dana-Farber Cancer Institute Molecular Biology Core Facilities prepared libraries from 2 ng of DNA, using the ThruPLEX DNA-seq kit (#R400427, Rubicon Genomics), according to the manufacturer's protocol; submitted the finished libraries to quality control analyses as described in the bulk RNA-seq Methods section; ChIP-seq libraries were uniquely indexed in equimolar ratios, and sequenced to a target depth of 40 M reads on an Illumina NextSeq500 run, with single-end 75 bp reads.

*mCRPC LuCaP PDXs.* ChIP-sequencing for AR (N-20; 6 µg, #sc-816, Santa Cruz Biotechnology), FOXA1 (4 µg, #ab23738, Abcam) and H3K27ac (1 µg, #C15410196, Diagenode), was performed at the Dana-Farber Cancer Institute using the protocol described previously[32,64].

*LNCaP MYC model.* Published ChIP-seq data (GSE73995[15]) was downloaded and reanalyzed.

**Bioinformatics analyses—ChIP-seq**
*Peak calling and data analysis.* All samples were processed through the computational pipeline developed at the Dana-Farber Cancer Institute Center for Functional Cancer Epigenetics (CFCE) using primarily open source programs. Raw Illumina output was converted to FASTQ format using Illumina Bcl2fastq (2.18). Sequence tags were aligned with Burrows-Wheeler Aligner (BWA; 0.7.17-r1188) to build mm9 or hg19 and uniquely mapped, non-redundant reads were retained[65]. These reads were used to generate binding sites with Model-Based Analysis of ChIP-seq 2 (MACS; 2.1.1.20160309), with a q-value (FDR) threshold of 0.01[66]. We evaluated multiple quality control criteria based on alignment information and peak quality: (i) sequence quality score; (ii) uniquely mappable reads (reads that can only map to one location in the genome); (iii) uniquely mappable locations (locations that can only be mapped by at least one read); (iv) peak overlap with Velcro regions, a comprehensive set of locations – also called consensus signal artifact regions – in the genome that have anomalous, unstructured high signal or read counts in next-generation sequencing experiments independent of cell line and of type of experiment; (v) number of total peaks (the minimum required was 1,000); (vi) high-confidence peaks (the number of peaks that are tenfold enriched over background); (vii) percentage overlap with known DHS sites derived from the ENCODE Project (the minimum required to meet the threshold was 80%); and (viii) peak conservation (a measure of sequence similarity across species based on the hypothesis that conserved sequences are more likely to be functional). Typically, if a sample fails one of these criteria, it will fail many (locations with low mappability will likely have low peak numbers, many of which will likely be in high-mappability regions, etc.).

*DNA binding motif analyses.* Peaks from each group were used for motif analysis by the motif search findMotifsGenome.pl in HOMER (3.0.0)[67], with cutoff q-value ≤ 1e-10.

*Sample-sample correlation and differential peaks analysis.* Sample-sample correlation and differential peaks analysis was performed by the CoBRA pipeline (2.0)[68]. Peaks from all samples were merged to create a union set of sites for each transcription factor and histone mark. Read densities were calculated for each peak for each sample, which were used for comparison of cistromes across samples. Sample similarity was determined by hierarchical clustering using the Spearman correlation between samples. Tissue-specific peaks were identified by DESeq2 (1.18.1) with adjusted $P ≤ 0.05$. Total number of reads in each sample was applied to size factor in DESeq2 (1.18.1), which can normalize the sequencing depth between samples.

*ChIP-seq profiles.* Given varying alignment of reads or fragments across samples, coverage track bigwig files were calculated for each sample that reflected the coverage signal and sequencing depth using the Chilin pipeline[69]. The deepTools (2.3.5) package computeMatrix further computed the average score for each of the samples. Finally, a profile heat map was created based on the scores at genomic positions within 2 kb upstream and downstream of the AR binding sites. All samples were ranked by the average score. ChIP-seq enrichment for transcription factors and histone marks at the loci of selected genes were visualized and plotted using karyoploteR R package (1.12.4)[70].

*RNA Pol II analysis.* RNA Pol II traveling ratio (TR) scores for each gene was calculated by comparing the ratio between RNA Pol II density in the promoter region and in the gene body region[42]. The promoter region was defined as −30 bp to +300 bp relative to the transcriptional start site (TSS) and the gene body as the remaining length of the gene. We calculated the bins per million mapped reads (BPM) use bamCoverage and computeMatrix in deepTools (2.3.5) for promoter and gene body regions. The TR difference between WT and MYC were calculated by TR value in WT minus TR value in MYC. Ranking plot of the WT - MYC TR difference for all Pol II bound genes revealed a clear point in the distribution of travel ratio difference where the difference began increasing/decreasing rapidly. To geometrically define this point, we found the x-axis point for which a line with a slope of 1 was tangent to the curve. We defined 246 genes above the increasing point to be pause release genes and 556 genes below the decreasing point to be the pause genes by MYC overexpression. DeepTools (2.3.5) function plotProfile and plotHeatmap were used to create the Pol II occupancy (the region ± 3 kb from the start and end of the gene) summary profiles and heatmaps. Kolmogorov-Smirnov test is applied to the TR distribution difference between WT and MYC for Hallmark Androgen_response genes.

**Epigenomics and transcriptomics integration.** All genes within the 100 kb of gained AR binding sites in MYC samples were selected, k-means clustering of 3 was applied. Cells were ordered by the pseudotime. GSEA analysis was done using the gene sets deposited in the GSEA website (https://www.gsea-msigdb.org/gsea/msigdb/annotate.jsp; 4.1.0). Binding and expression target analysis (BETA; 1.0.7) was used to integrates ChIP-seq of transcription factors with differential gene expression data and infer the dysregulated genes[40].

**Prostate cancer clinical datasets analyses**
*The Cancer Genome Atlas (TCGA).* RNA-seq readcount and clinical data from 488 samples with prostate cancer (PRAD) were downloaded from the Cancer Genome Atlas (TCGA) database (https://cancergenome.nih.gov/) using Bioconductor package TCGAbiolinks (2.14.1)[71]. To calculate transcriptional signature scores, RNA-seq data was normalized to sequencing depth and TPM transformed. Hallmark Androgen_response and Hallmark MYC_targets_V1 gene sets were downloaded from MSigDB[72]. The AR-A signature comprising nine canonical AR transcriptional targets (*KLK3, KLK2, FKBP5, STEAP1, STEAP2, PPAP2A, RAB3B, ACSL3, NKX3-1*) was derived from previous published work[34]. Transcriptional signature scores were computed for every patient based on a non-parametric, rank-based method implemented in singscore R package (1.6.0)[73]. TCGA patients were assigned to the low or high group according to the cut-off point estimated by maximally selected rank statistic maxstat R package (0.7–25) of each signature[74]. Survival analysis was conducted using survival R package (3.2-3)[75], Kaplan-Meier were plotted using survminer R package (0.4.8)[76] and log-rank test was used to evaluate the overall statistical significance as well as the comparison between groups. Benjamini-Hochberg was used to correct for multiple testing.

*Validation cohort.* The META855 cohort containing 855 patients treated with radical prostatectomy with available transcriptomic, clinicopathological, and outcomes data selected from five published studies of the Decipher prostate genomic classifier test as previously described[35]. Microarray expression levels were normalized using the SCAN algorithm (SCAN.UPC R package; 2.28.0)[77]. The combination of the Hallmark Androgen_response/Hallmark MYC_targets_V1 and AR-A / Hallmark MYC_targets_V1 signatures and their association with BCR and metastatic progression was examined in the META855 cohort using the thresholds

obtained from quantiles defined in the TCGA dataset. Patients were divided in four groups and Kaplan-Meier analysis and log-rank test were conducted to evaluate differences in biochemical recurrence and metastatic progression. The prognostic association between the signatures and the clinicopathological factors was assessed using Cox proportional hazard modeling.

*Castration-resistant prostate cancer.* Published gene expression data (GSE126078[37]) was downloaded and data analysis was performed using VIPER (2.0)[58].

*Metastatic castration-resistant prostate cancer.* The SU2C International Dream Team cohort contains 429 mCRPC patients treated with a first-line ARSI (*i.e.* abiraterone acetate or enzalutamide)[39]. Patients underwent biopsy for the collection of mCRPC tissue and a total of 75 patients had matching transcriptomic profiling (RNA-seq) and outcomes data. Hallmark Androgen_response (missing gene expression data for *HERC3*) and Hallmark MYC_targets_V1 (missing gene expression data for *PRPF31*) transcriptional signature scores were computed for every patient based on a non-parametric, rank-based method implemented in singscore R package (1.6.0)[73] using gene expression as TPM (transcripts per million reads). Patients were assigned to the low or high group according to the cut-off point estimated by maximally selected rank statistic maxstat R package (0.7–25) of each signature[74]. Survival analysis was conducted using survival R package (3.2-3)[75], Kaplan-Meier were plotted using survminer R package (0.4.8)[76] and log-rank test was used to evaluate the differences in overall survival. The prognostic association between the signatures was assessed using Cox proportional hazard modeling.

**Reporting summary.** Further information on research design is available in the Nature Research Reporting Summary linked to this article.

## Data availability
The murine (bulk RNA-seq, scRNA-seq and ChIP-seq) and LuCaP PDXs (ChIP-seq) sequencing data reported in this paper were deposited on NCBI Gene Expression Omnibus (GEO) and are accessible through GEO Series accession number GSE163146 and GSE163220, respectively. The CRPC (bulk RNA-seq) and the LNCaP MYC model (microarray and ChIP-seq) publicly available data used in this study are available through GEO Series accession number GSE126078 and GSE73995, respectively. The remaining data are available within the Article, Supplementary Information or Source Data file provided with this paper.

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

## Acknowledgements

We thank Zach Herbert for technical assistance, Noriko Uetani for Figs. 1a, 3f and 8b design and drawings and Marie-Claude Gingras and Livia Garzia for critical review of this manuscript. T.H. is the recipient of the 100 Days Across Canada Bursary Award. J.L. is a recipient of a Canadian Institute of Health Research Frederick Banting and Charles Best Canada Graduate Scholarship-Master's and of a Research Institute of the McGill University Health Centre M.Sc. Studentship award. Establishment and characterization of the LuCaP PDX models has been supported by the Pacific Northwest Prostate Cancer SPORE (P50CA97186), the U.S. Department of Defense Prostate Cancer Biorepository Network (W81XWH-14-2-0183), the Prostate Cancer Foundation, the Institute for Prostate Cancer Research, and the Richard M. Lucas Foundation. We would like to thank the patients who generously donated tissue that made this research possible. G.Z. is a recipient of an Idea Development Award from the U.S. Department of Defense (PC150263) and the Barr Award from the Dana-Farber Cancer Institute. This work has been supported by National Institutes of Health grants to K.W.W. (R01 CA238039; R01 CA251599), a Prostate Cancer Foundation Challenge Award to M.M.P. and M.L.F. and grants to M.L.F. (National Institutes of Health, R01 GM107427, R01 CA251555 and R01 CA193910; U.S. Department of Defense, W81XWH-19-1-0565 and W81XWH-21-1-0234; the H.L. Snyder Medical Research Foundation; the Donahue Family Fund; the Mayer Foundation; the Cutler Family Fund for Prevention and Early Detection; the Claudia Adams Barr Program for Innovative Cancer Research). E.C., M.B. and H.W.L. acknowledge support from the National Institutes of Health (P01 CA163227-06A1). D.P.L. is a William Dawson Scholar of McGill University, a Lewis Katz – Young Investigator of the Prostate Cancer Foundation, the recipient of a Scholarship for the Next Generation of Scientists from the Cancer Research Society and is also a Research Scholar – Junior 1 from The Fonds de Recherche du Québec – Santé. The work reported here was funded by a Canadian Institutes of Health Research project grant (PJT-162246) to D.P.L.

## Author contributions

Conceptualization, X.Q., M.B., H.W.L. and D.P.L.; Methodology: X.Q., N.B., A.F. and Y.X.; Software: X.Q., N.B., A.F. and Y.X.; Validation: Y.L., E.D. and D.E.S.; Formal Analysis, X.Q., N.B., A.F., Y.X., S.G., Q.T., Y.Z. and D.P.L.; Investigation, T.H., A.d.P., A.M.L., W.A., J.L., G.Z., S.S., J-H.S., C.B., E.O'C., P.C. and D.P.L.; Resources, E.M.S., R.J.K., S.W., C.M., L.E., M.L., K.W.W., M.M.P., E.C., M.L.F., X.S.L., M.B., H.W.L. and D.P.L.; Data Curation, X.Q. and N.B.; Writing – Original Draft, D.P.L.; Writing – Review & Editing, X.Q., N.B., G.Z., E.C., M.F., H.W.L. and D.P.L.; Visualization, X.Q., N.B. and D.P.L.; Supervision, H.W.L. and D.P.L.; Project Administration, H.W.L. and D.P.L.; Funding Acquisition, M.B., H.W.L. and D.P.L.

## Competing interests

R.J.K. receive royalties from GenomeDx (now Veracyte) for Decipher testing. S.W. receives research funding from PreludeDX. K.W.W. serves on the scientific advisory board of T-Scan Therapeutics, SQZ Biotech, Nextechinvest and receives sponsored research funding from Novartis. He is a co-founder of Immunitas, a biotech company. These activities are not related to the research reported in this publication. D.E.S. receives personal fees from Janssen, AstraZeneca, and Blue Earth and funding from Janssen. E.C. received research funding under institutional SRA from Janssen Research and Development, Bayer Pharmaceuticals, KronosBio, Forma Pharmaceutics, Foghorn, Gilead, Sanofi, AbbVie, MacroGenics, and GSK. M.L.F. reports other support from Nuscan Diagnostics outside the submitted work. X.S.L. conducted the work while being a faculty at the Dana-Farber Cancer Institute and is currently a board member and CEO of GV20. M.B. and H.W.L. receives sponsored research support from Novartis. M.B. is a consultant to Aleta Biotherapeutics and H3 Biomedicine and serves on the SAB of Kronos Bio. The remaining authors declare no competing interests.

## Additional information

[1]Center for Functional Cancer Epigenetics, Dana-Farber Cancer Institute, Boston, MA, USA. [2]Department of Medical Oncology, Dana-Farber Cancer Institute, Harvard Medical School, Boston, MA, USA. [3]Cancer Research Program, Research Institute of the McGill University Health Centre, Montréal, QC, Canada. [4]Department of Anatomy and Cell Biology, McGill University, Montréal, QC, Canada. [5]Division of Urology, Department of Surgery, McGill University, Montréal, QC, Canada. [6]Department of Cancer Immunology and Virology, Dana-Farber Cancer Institute, Harvard Medical School, Boston, MA, USA. [7]Division of Experimental Medicine, Department of Medicine, McGill University, Montréal, QC, Canada. [8]Departments of Oncologic Pathology and Pathology, Dana-Farber Cancer Institute and Brigham's Women Hospital, Boston, MA, USA. [9]Institute of Molecular Genetics, National Research Council, Pavia, Italy. [10]Department of Data Science, Dana-Farber Cancer Institute, Harvard T.H. Chan School of Public Health, Boston, MA, USA. [11]Decipher Biosciences, San Diego, CA, USA. [12]Department of Urology, Northwestern University, Chicago, IL, USA. [13]Department of Urology, Mayo Clinic, Rochester, MN, USA. [14]Center for Health Research, Kaiser Permanente Northwest, Portland, OR, USA. [15]Department of Urology, University of Washington, Seattle, WA, USA. [16]Division of Medical Oncology, Department of Medicine, Cedars-Sinai Medical Center, Los Angeles, CA, USA. [17]Cedars-Sinai Samuel Oschin Comprehensive Cancer Institute, Los Angeles, CA, USA. [18]Center for Bioinformatics and Functional Genomics, Department of Biomedical Sciences, Cedars-Sinai Medical Center, Los Angeles, CA, USA. [19]Department of Pathology and Laboratory Medicine, Weil Cornell Medicine, New York Presbyterian-Weill Cornell Campus, New York, NY, USA. [20]Department of Radiation Oncology, University Hospitals Seidman Cancer Center, Case Western Reserve University School of Medicine, Cleveland, OH, USA. [21]The Eli and Edythe L. Broad Institute, Cambridge, MA, USA. [22]These authors contributed equally: Xintao Qiu, Nadia Boufaied. [23]These authors jointly supervised this work: Henry W. Long, David P. Labbé. ✉email: henry_long@dfci.harvard.edu; david.labbe@mcgill.ca

