## [Peer Review File · Nature Communications]

MYC drives aggressive prostate cancer by disrupting transcriptional pause release at androgen receptor targetsReviewers' Comments:

Reviewer #1:

Remarks to the Author:

Previous studies have demonstrated that MYC is a major driver for prostate cancer. In this study, Qiu et al. found that MYC overexpression reduces AR signaling in luminal prostate cells without affecting AR expression. They further demonstrated that MYC overexpression disrupts Pol II pausing release at AR-regulated genes without affecting binding of AR transcription complex at the AR enhancers. Overall, this is a very interesting and worthy area of research. Several concerns should be addressed to improve the manuscript.

1. The authors have previously found that the murine AP is mostly unaffected by MYC overexpression while PIN penetrance reaches 83% and 97% in the DLP and VP, respectively. Based on their observation that hgMYC transgene expression is only prevalently detected in the VP lobe, the authors hypothesize that there is a "paracrine transcriptional reprogramming upon MYC overexpression and prostate transformation". Is there any evidence to support this hypothesis? Also why PIN penetrance reaches 83% in DLP but not in AP?
2. Line 125-129: could the authors explain the rationale for overexpression of hg19MYC rather than mm10Myc in the mouse model? Why the authors choose ARR2Pb to drive hg19MYC? Why hg19MYC expression is more prevalent in the VP lobe compared with the AP and DLP lobes?
3. Figure 1H and 1I: The expression of hg19MYC is much higher in VP than in DLP, but why the PIN penetrance in DLP and VP is comparable?
4. For the 8 PDXs used in this study: are high expression of MYC associated with more aggressive phenotype of PDX models?
5. Figure 7E: why there are two bands for Probasin? Could the authors also show anti-MYC blots?
6. Figure 7J-Q: why MYC promotes Pol II pausing release at some genes (e.g. MYC target genes) but inhibits Pol II pausing release at other genes (e.g. AR target genes)? Could the authors discuss these findings?

Reviewer #2:

Remarks to the Author:

MYC is known to be give cancer in various murine tissues when expressed as a transgene, as such the results presented in this work, that MYC over-expression is leading to disease recurrence in castrated rodents is not surprising, nor unexpected. Likewise, others have shown that MYC over-expression deregulates the AR transcriptional program (see for example, REF 15). Perhaps the main finding of this manuscript could be considered the "transcriptional pausing" but I find this explanation questionable and would argue that biochemical assay sensitivity is not sufficient to claim that pausing effect occurs. I believe that the experimental approach chosen is not sufficient to address transcriptional pausing, and other effects can play a role. For example, rigorous experiments in previous work (Cell. 2012 Sep 28; 151(1): 56-67), showed that MYC "increased transcription elongation by RNA polymerase II (RNA Pol II) and increased levels of transcripts per cell." Therefore, looking through the existing literature reports, unfortunately I have to restrain from recommending this work for publishing at Nature Communications journal. With all due respect, I could not find new biology, or insights that would be impossible to make from the already published reports.

Other concerns:

Authors claim:

L91-94: We determine that an active MYC transcriptional program and low AR activity identify prostate cancer patients predisposed to fail standard-of-care therapies and most likely to develop metastatic castration resistant prostate cancer (mCRPC).

L94-96: Accordingly, we found that high MYC mRNA expression in castration-resistant tumors is also

associated with a weakened canonical AR transcriptional program and a repurposing of the AR cistrome.
However, both of these claims have been shown previously (see REF 15).

Likewise: claim in L99-100, has been already shown by others (REF 15)

Please comment in the main text why you have not recovered B cells, NK cells, Dendritic cells that are known to be present in prostate tissue. Also please comment why seminal vesicle epithelium is missing in your tSNE plot?

The color choice in Figure S2-4 is poor, please choose different kind color bar.

Figure S5 - what this curve suppose to mean?

L233: The difference between AR_low/MYC_high vs AR_low/MYC_low is too small to reliable claim that AR-A tumors with high MYC expression are associated with a faster occurrence of BCR. In other words, the MYC effect on AR-A seems to be context dependent and thus there is not enough experimental evidence to support the claim that MYC_high / AR-A low leads to BCR faster.

Looking at Figure 5F statistics is seems that differences between the two groups (AR_low/MYC_high and AR_low/MYC_low) are not statistically significant., P-values are high. Therefore, I would argue that your conclusion that high_MYC / low_AR modulates disease progression is not sufficiently substantiated at this point. Perhaps you can clarify that in the text for future readers..

Is the functional role of MYC cell type-specific? if so, please comment that in the text when talking about AP, DLP, VP lobes and other regions of prostate.

Line 130: Perhaps this is just semantics but I do not see "a robust and uniform MYC-driven PIN transition" in DLP case, presented in Figure 1I.

Lines 145-147. I find following claim confusing: "a state that was independent of human or murine MYC transcript level (Figure 2D)".

Looking at Figure 2D is seems that there are differences in expression (when comparing Lum1/2 vs LumMYC/LumMYC high, distributions).

Line 150: "Taken together, these results demonstrate that MYC-driven transcriptional reprogramming can be readily captured by single-cell transcriptomics". I think such conclusion is just too obvious. scRNA-Seq has advances far beyond this point.

Line 182: "... these results indicate that AR-transcriptional program is compromised upon MYC overexpression." Again, perhaps this is just semantics, but if MYC overexpression suppresses or outcompetes the AR-activated genes, because of shared cis elements, it would incorrect to claim that "AR-transcriptional program is compromised".

Material and Methods

The dissociation protocol seems to be very long (>2h hours), a time during which cells are expected to experience stress and change their transcriptional outcome drastically. Investigating subtle effects such as transcriptional pausing on sample that were dissociated for 2 hours rises questions about introduction of technical biases. I recognized that my request might be stretched too far but to rule out biological and technical artifacts I think you would need alternative confirmation that transcriptional pausing is actually happening, for example on cells that are not being dissociated for such a long time (e.g. using nuc-seq).

Reviewer #3:

Remarks to the Author:

The manuscript by Qiu et al. describes the crosstalk between MYC expression androgen receptor signaling in the mouse prostate. Authors compared the transcriptomes single cells derived from lobes of the three prostate lobes, the anterior, dorsolateral and ventral prostate. In addition, authors compared the transcriptomes of the lobes of probasin-MYC transgenic mice driving an overexpression of the MYC oncogene in the prostate leading to PIN. The data suggest a repression of androgen receptor (AR) signaling in the lobes affected by MYC overexpression without reducing AR gene expression itself. Furthermore, ChIP-seq with primary tissues were performed, integrating AR, FOXA1 and H3K27ac ChIP-seq to single-cell transcriptomes. TCGA datasets indicates a correlation among human PCa characterized by a low AR transcriptional signature with concurrent high MYC transcriptional signature to be associated with a short time to biochemical recurrence whereas high AR in combination with low MYC signature associates with the longest time to biochemical recurrence. Gene expression profiling from 59 AR-positive CRPC tumors revealed that AR activity is negatively correlated with MYC expression supporting that MYC inhibits the canonical AR transcriptional program in CRPC.

LuCaP patient-derived xenografts were also used to analyze cistrome and FOXOA1 occupancy. A greater FOXA1 occupancy was observed at AR gained binding sites in MYC-high compared to low MYC. Functionally these observations were confirmed by overexpressing MYC in the mouse model system. AR binding was significantly associated with genes downregulated by MYC overexpression. However interestingly, AR binding without inactivation of chromatin nearby androgen response genes remained largely unchanged following MYC overexpression as well as MYC binding nearby MYC target genes also remained unchanged by MYC overexpression despite MYC inhibits AR transcriptome signature. Authors suggest mechanistically, shown by ChIP-seq of Pol II, that Pol II pauses at promoters upon MYC overexpression.

Since these patterns suggest a MYC driven altered ratio of initiating and elongating RNA Pol II at AR-regulated genes.

Authors have impressively shown in native tissue the crosstalk between MYC and AR in in vivo mouse model and human PCa cell derived xenografts. Furthermore, the mechanism suggested is that MYC overexpression leads to pausing of Pol II in a subset of androgen responsive genes.

Minor points:

- Please indicate how many fresh prostate lobes were used in the bulk RNA seq?
- On Page 14 it is mentioned tyrosinization. In which step that enzyme was used?
- Fig. 4H and Fig. 7K: GSEA analysis indicates that the p53 pathway is involved in both MYC dependent increase and decrease.

Although this is bioinformatically possible, the request would be to analyze the same data in addition with an IPA (Ingenuity Pathways Analysis).

Reviewer #4:

Remarks to the Author:

Qiu et al.,

Qiu et al. describe that MYC overexpression in prostate cancer cells hinders the release of paused RNA Pol2 at androgen receptor (AR) target genes, thereby reducing the transcriptional output of AR-driven transcriptional programs. This transcriptional shift leads to a more aggressive phenotype of prostate cancer cells. While most of the data support this hypothesis and are scientifically sound, the data structure of the manuscript and its presentation in the figures is partially confusing and to some

extent hinders a clear understanding of the principal findings. Deconvolution of the data reveals limited novelty in comparison to the findings of Barfeld et al., (PMID 28412251) already demonstrating the AR-antagonizing role of MYC in prostate cancer. In addition, there is apparently some controversy in the field with respect to the role of MYC as there is no effect on AR as shown in a mouse model (Kim et al. Oncogene 2012) and in cell lines models AR knockdown has been shown to reduce the expression of MYC (Gao et al. Plos One 2013). The present study mechanistically expands this observation by demonstrating MYC-induced hinderance of RNA Pol2 pause release at AR target genes. To increase the novelty of the study, ideally, this finding should be followed up mechanistically in more detail to justify publication in this journal. As is, due to lack of novelty, and limited functional and translational follow-up the reviewer cannot recommend publication of the manuscript in its present form.

General comments:

1. For increased clarity Figures 1, 2 and 3 should be combined into one Figure that is condensed to principal findings of MYC activation influencing transcription in the mouse model by diminishing AR transcriptional output. Especially Figure 1B/C/D, Figure 2 and the second part of Figure 3 are not very intuitive and add little to the overall finding.
2. Mechanistically, it would be interesting to answer whether increased MYC binding at AR target sites is dependent on MYC binding to canonical E-Box sequence motifs or not. This is interesting to understand MYC-induced gene repression and whether it differs from MYC-induced gene induction on a DNA binding sequence level.
3. What is the translational impact of these results? The authors should present data that would hint at a possible therapeutic intervention that would harness the findings either in the utilized mouse model or in cellular models. This type of data may justify the publication in this journal.

Minor points:

1. Fig 1D/E/F the authors should comment / offer an explanation why high MYC does not alter transcription in the AP lobe.
2. Fig 1H/I the authors claim PIN transformation is MYC-driven. However, PIN transformation in the DLP is comparable to VP (Fig1H - no statistical difference indicated) despite lower MYC levels and 10x lower number of luminal cells in the DLP vs VP. This contradiction must be explained. Otherwise, the claim that PIN is MYC-driven cannot be made.
3. Fig S4 the figure is hardly comprehensible due to the very poor contrast and expression differences are not clearly visible – this must be displayed more clearly.
4. Fig 2B/3A: The two Luminal MYC subpopulations do not differ in MYC expression, however, only a minority is “highly proliferating”. It would be interesting to include this distinction in subpopulation also in the gene set analysis in Fig 3A and evaluate the contribution of either luminal subpopulation to the overall luminal transcriptional profile. Given the possible different origin of these subpopulations and comparable MYC expression but differential aggressiveness, it would be interesting to evaluate how and why comparable MYC levels lead to differential transcriptional output.

REVIEWER COMMENTS

Reviewer #1

Expert in ChIP-seq and RNA-seq analysis

Remarks to the Author

Previous studies have demonstrated that MYC is a major driver for prostate cancer. In this study, Qiu et al. found that MYC overexpression reduces AR signaling in luminal prostate cells without affecting AR expression. They further demonstrated that MYC overexpression disrupts Pol II pausing release at AR-regulated genes without affecting binding of AR transcription complex at the AR enhancers. Overall, this is a very interesting and worthy area of research. Several concerns should be addressed to improve the manuscript.

Response: We thank the reviewer for the helpful and supportive comments.

1. The authors have previously found that the murine AP is mostly unaffected by MYC overexpression while PIN penetrance reaches 83% and 97% in the DLP and VP, respectively. Based on their observation that hgMYC transgene expression is only prevalently detected in the VP lobe, the authors hypothesize that there is a “paracrine transcriptional reprogramming upon MYC overexpression and prostate transformation”. Is there any evidence to support this hypothesis? Also why PIN penetrance reaches 83% in DLP but not in AP?

Response: Based on our scRNA-seq analysis, the human MYC transgene is predominantly expressed in luminal cells (**Figure 2C**), which results in the robust transcriptional reprogramming of this cell subpopulation (**Figure 3A**). Importantly, although only a limited proportion of basal cells (18.3%; **Figure 2C**) express the human MYC transgene, we did observe a significant transcriptional reprogramming in the basal cell subpopulation (**Figure 3A**). Hence, we hypothesized that basal cell transcriptional reprogramming might be partly due to a MYC-dependent paracrine signaling originating from the luminal cell subpopulation. Additional experiments will be required to validate this hypothesis but are outside the scope of this manuscript. The marginal prostate intraepithelial neoplasia (PIN) penetrance observed in the AP lobe is most likely due to the weak expression of the human MYC transgene in this lobe (**Figure 1I**) as the level of murine prostate transformation was shown to be proportional to the strength of the human MYC transgene expression ¹.

2. Line 125-129: could the authors explain the rationale for overexpression of hg19MYC rather than mm10Myc in the mouse model? Why the authors choose ARR2Pb to drive hg19MYC? Why hg19MYC expression is more prevalent in the VP lobe compared with the AP and DLP lobes?

Response: The Hi-MYC genetically engineered mouse model (GEMM) has been developed in 2003 by the group of Charles Sawyers with the goal of determining the consequence of increased human *c-MYC* in the prostate ¹. Therefore, by choosing to overexpress *hg19MYC* rather than *mm10Myc*, one could argue that this increases the translational relevance of this prostate cancer model. The ARR₂/probasin promoter was chosen to ensure high transgene expression levels in the mouse prostatic epithelium. However, the penetrance of ARR₂/probasin-regulated gene expression across murine prostate lobes is variable, with greater expression in VP and DLP compared to the AP, as previously described ².

3. *Figure 1H and II: The expression of hg19MYC is much higher in VP than in DLP, but why the PIN penetrance in DLP and VP is comparable?*

Response: The reviewer raises an interesting point. Prostate intraepithelial neoplasia (PIN) is a premalignant condition that precedes the onset of invasive adenocarcinoma in both in human and the Hi-MYC genetically engineered mouse model (GEMM) of prostate cancer^{1, 3}. MYC overexpression results in phenotypes ranging from PIN to overt adenocarcinoma. However, the phenotype primarily depends on the “level” of MYC expression. Along this line, the Lo-MYC GEMM of prostate cancer only develop PIN after a period of high latency while the Hi-MYC GEMM of prostate cancer progresses to the adenocarcinoma stage¹. Therefore, although MYC levels in the DLP are sufficient in leading to comparable PIN penetrance to the VP of 12-week-old animals, it may be that in younger animals, PIN penetrance in the DLP might be inferior to the VP due to a weaker *MYC* transgene expression.

4. *For the 8 PDXs used in this study: are high expression of MYC associated with more aggressive phenotype of PDX models?*

Response: The growth of the LuCaP PDX models is heterogenous and our historical data on their growth do not show consistent/significant differences between the high and low MYC PDX models. However, Gene Set Enrichment Analysis (GSEA) showed that PDX models with high MYC expression are associated with an enrichment in MYC-transcriptional activity (MYC_targets_V1) and in the E2F targets gene set (E2F_targets), which is indicative of more proliferative tumors (**Reviewer Figure 1**).

Reviewer Figure 1: Gene Sets Enrichment Analysis (GSEA; Hallmark, $P < 0.05$) showed that *MYC*-high LuCaP mCRPC PDX models are associated with an enrichment in MYC-transcriptional activity (MYC_targets_V1) and in the E2F targets gene set (E2F_targets) compared to *MYC*-low LuCaP mCRPC models.

5. *Figure 7E: why there are two bands for Probasin? Could the authors also show anti-MYC blots?*

Response: According to the antibody datasheet (#sc-393830, Santa Cruz Biotechnology), the expected molecular weight of probasin is 22 kDa, corresponding to the lower band in *Figure 7E*. The upper band is most likely unspecific to Probasin since it is unaltered following human *MYC* transgene expression although *Pbsn* expression levels are significantly abrogated ($P = 5.88e-95$; $FDR = 1.02e-90$; **Figure 7F**). The corresponding anti-MYC blot from the same animals is presented in **Figure 3E**.

6. *Figure 7J-Q: why MYC promotes Pol II pausing release at some genes (e.g. MYC target genes) but inhibits Pol II pausing release at other genes (e.g. AR target genes)? Could the authors discuss these findings?*

Response: Given that increased MYC expression lead in greater RNA Pol II promoter-proximal pausing at genes regulated specifically by the AR without an accompanying deactivation of AR-bound enhancers, these results support cofactor redistribution as a potential mechanism for MYC-mediated transcriptional repression. This is an active area of research for our laboratory and we are currently investigating potential cofactor candidates that could be mediating RNA Pol II promoter-proximal pausing at AR-regulated genes following MYC overexpression.

Reviewer #2

Expert in sc-RNAseq analysis

Remarks to the Author

MYC is known to give cancer in various murine tissues when expressed as a transgene, as such the results presented in this work, that MYC over-expression is leading to disease recurrence in castrated rodents is not surprising, nor unexpected. Likewise, others have shown that MYC over-expression deregulates the AR transcriptional program (see for example, REF 15). Perhaps the main finding of this manuscript could be considered the "transcriptional pausing" but I find this explanation questionable and would argue that biochemical assay sensitivity is not sufficient to claim that pausing effect occurs. I believe that the experimental approach chosen is not sufficient to address transcriptional pausing, and other effects can play a role. For example, rigorous experiments in previous work (Cell. 2012 Sep 28; 151(1): 56–67), showed that MYC "increased transcription elongation by RNA polymerase II (RNA Pol II) and increased levels of transcripts per cell." Therefore, looking through the existing literature reports, unfortunately I have to restrain from recommending this work for publishing at Nature Communications journal. With all due respect, I could not find new biology, or insights that would be impossible to make from the already published reports.

Response: In their 2012 *Cell* manuscript, Lin *et al.* provided groundbreaking evidence suggesting that c-MYC act as a transcriptional amplifier of the cell's gene expression program⁴. Two subsequent manuscripts, published back-to-back in *Nature*, argued that transcriptional amplification co-exists with selective up- and down-regulation of specific MYC target genes^{5,6}. Another report recently published in *Science* rather suggest that MYC primarily acts as a selective transcriptional activator controlling metabolic processes such as ribosome biogenesis and *de novo* purine synthesis⁷. In the context of our *in vivo* experiments using the MYC-driven GEMM of prostate cancer, our data provide robust evidence that MYC overexpression antagonizes the canonical AR transcriptional program. It is important to highlight that our experiments were not designed to determine whether MYC acts as a transcriptional amplifier in the context of prostate cancer.

However, our data revealed that MYC-driven repression of the AR transcriptional program is not associated with the disengagement of AR or the loss of the H3K27ac mark. Rather, we observed greater RNA Pol II promoter-proximal pausing and non-productive transcription at AR-dependent genes repressed by MYC *in vivo*. While it is currently thought that MYC overexpression drives addiction into transcriptional elongations in solid tumors (reviewed in Chen *et al.*)⁸, our novel work suggests that MYC-driven prostate cancer initiation and progression also rely on the RNA Pol II promoter-proximal pausing for transcriptional repression at genes regulated specifically by the AR. We have contextualized the novelty of our work with regards to the Barfeld *et al.* manuscript (REF 15; regarding MYC and AR interplay, clinical relevance and prostate cancer initiation)⁹ and clarified our Pol II ChIP-seq workflow used to quantify transcriptional pause release at AR-regulated genes (which used flash-frozen murine prostate lobes as starting material, ruling out potential biological and technical artifacts that could have been associated with a dissociation protocol) below. We hope that altogether, we have addressed the reviewer's concerns.

Other concerns

Authors claim:

L91-94: *We determine that an active MYC transcriptional program and low AR activity identify prostate cancer patients predisposed to fail standard-of-care therapies and most likely to develop metastatic castration resistant prostate cancer (mCRPC).*

L94-96: *Accordingly, we found that high MYC mRNA expression in castration-resistant tumors is also associated with a weakened canonical AR transcriptional program and a repurposing of the AR cistrome.*

However, both of these claims have been shown previously (see REF 15).

Likewise: claim in L99-100, has been already shown by others (REF 15).

Response: The reviewer raises an important point and we would like to take this opportunity to discuss the Barfeld *et al.* (REF 15; *EBioMedicine* 18 (2017) 83-93)⁹ manuscript and contextualize our findings as well as their novelty.

MYC and AR interplay (L94-96): Although MYC and AR are central transcription factors in prostate cancer etiology, a very limited number of studies provided insight into their interplay. We acknowledge that Barfeld *et al.* previously reported an antagonistic role of MYC overexpression on AR transcriptional activity, but the molecular basis for this interaction and its physiological significance have remained obscure. Along this line, our study complements the Barfeld *et al.* manuscript, which was solely based on *in vitro* models (LNCaP and VCaP prostate cancer cell lines), by demonstrating at a single-cell level the antagonistic impact of MYC overexpression on the AR transcriptional program *in vivo*. Importantly, our *in vivo* study provides for the first time a clear mechanistic insight into the MYC-driven repression of the AR transcriptional program through increased RNA polymerase II (Pol II) promoter-proximal pausing and non-productive transcription at AR-dependent genes.

Clinical relevance (L91-94): In their study, Barfeld *et al.* identified 166 genes induced by the AR but antagonized by MYC overexpression and then tested the individual prognostic value of genes in two published datasets, without considering either MYC status or MYC transcriptional activity. In the Glinsky cohort, the high expression of 26 genes were found associated with a longer time to biochemical recurrence (BCR), 10 genes with a shorter time to BCR and 130 genes not significantly affected by disease progression. Similarly, in the Taylor cohort, the high expression of 46 genes were found associated with longer time to biochemical recurrence (BCR), 10 genes with a shorter time to BCR and 110 genes not significantly affected by disease progression. Altogether, these confusing results revealed that taken individually, AR-regulated genes antagonized by MYC overexpression are not robust prognostic biomarkers. Based on our *in vivo* evidence, we reasoned that a holistic approach, capturing the status of both the AR and MYC transcriptional program, is necessary to determine the ongoing transcriptional rewiring at a given time and disease aggressiveness. We tested this hypothesis in the TCGA cohort and validated our findings in the Spratt *et al.* cohort using the thresholds obtained from quantiles defined in the TCGA dataset. Therefore, our conclusion that an active MYC transcriptional program and low AR activity identify prostate cancer patients predisposed to fail standard-of-care therapies and most likely to develop metastatic castration resistant prostate cancer (mCRPC) is **novel**. Critically, Barfeld *et al.* did not interrogate the mCRPC transcriptional program or AR cistrome since their studies solely used *in vitro*, castration sensitive model (LNCaP) for transcriptomics and epigenomics experiments. Thus, our conclusion that high MYC mRNA expression in castration-resistant tumors is also associated with a weakened canonical AR transcriptional program and a repurposing of the AR cistrome is **novel**.

Prostate cancer initiation (L99-100): Again, since Barfeld *et al.* exclusively focus on the LNCaP prostate cancer cell line for transcriptomics and epigenomics experiments, this study did not provide information regarding the impact of MYC overexpression on tumor initiation. By leveraging the Hi-MYC GEMM and wild-type animals, our studies provide a ***novel*** insight into prostate cancer initiation driven by MYC overexpression and its impact of the AR transcriptional program.

Please comment in the main text why you have not recovered B cells, NK cells, Dendritic cells that are known to be present in prostate tissue. Also please comment why seminal vesicle epithelium is missing in your tSNE plot?

Response: For scRNA-seq, we dissected each mouse prostate lobes (AP, DLP, VP) and immediately processed the tissues for single-cell transcriptomics. Importantly, we did not enrich for immune cell subpopulations (*e.g.* via CD45+ enrichment). Therefore, only a limited number of immune cells, proportional to their relative abundance, was captured in our samples (mostly the macrophage/monocyte and T cell subpopulations). Regarding the seminal vesicle, it has been dissected but discarded since it is not a prostatic tissue and because this tissue remains unaffected in the Hi-MYC GEMM due to the lack of MYC transgene expression.

The color choice in Figure S2-4 is poor, please choose different kind color bar.

Response: We agree with the Reviewer and have increased the size of each cell represented on the tSNE plots between 200% (**Figure 1, Figure 2, Figure S3**) and 400% (**Figure S2, Figure S4, Figure S5**) to improve readability.

Figure S5 - what this curve suppose to mean?

Response: This is a graphical representation of the slingshot pseudotime inference used for ordering of luminal cells (**Figure 4G**) in an unbiased fashion (starting from luminal 2, luminal 1, luminal MYC and luminal MYC high. prolifer.) as described by Street *et al.* ¹⁰. Briefly, this tree method, which was ranked the best in a recent comparison of 45 single-cell trajectory inference methods ¹¹, uses pre-existing clusters to infer lineage hierarchies (based on minimal spanning tree, MST) and align cells in each cluster on a pseudotime trajectory. We performed slingshot pseudotime inference on combined WT and MYC VP samples. T-distributed Stochastic Neighbor Embedding (t-SNE) is applied to visualize cells in a 2-D space based on the principal components after dimensionality reduction. Slingshot analysis was based on the t-SNE coordinates.

L233: The difference between AR_low/MYC_high vs AR_low/MYC_low is too small to reliable claim that AR-A tumors with high MYC expression are associated with a faster occurrence of BCR. In other words, the MYC effect on AR-A seems to be context dependent and thus there is not enough experimental evidence to support the claim that MYC_high / AR-A low leads to BCR faster.

Response: We disagree with the reviewer since low AR-A tumors with concurrent high MYC transcriptional signature (AR_low/MYC_high) are associated with a faster time to BCR compared to AR_low/MYC_low tumors according to log-rank tests in the *discovery cohort* (TCGA cohort; **Figure 5B**) and univariable analysis in the *validation cohort* (Spratt *et al.*; **Figure 5D**). Strikingly, Kaplan-Meier curves, univariable and multivariable analyses revealed that patients with tumors harboring an AR_low/MYC_high signature were the most likely to develop metastatic disease (**Figure 5E-F**). Altogether, our results strongly suggest that concurrent AR_low/MYC_high transcriptional signatures identify a subgroup of patients that are predisposed to fail standard-of-care therapies and progress to develop metastatic disease.

Looking at Figure 5F statistics it seems that differences between the two groups (AR low/MYC high and AR low/MYC low) are not statistically significant., P-values are high. Therefore, I would argue that your conclusion that high_MYC / low_AR modulates disease progression is not sufficiently substantiated at this point. Perhaps you can clarify that in the text for future readers.

Response: We are confused by this comment. In Figure 5F, AR_low/MYC_low tumors are the reference signature. Univariable analyses revealed that AR_low/MYC_high are significantly associated with an increased risk to develop metastatic disease (HR = 2.93, 95% CI 1.68-5.10; $P < 0.001$) compared to AR_low/MYC_low tumors. Moreover, this finding remained significant in a multivariable competing risks regression analysis adjusting for age, prostate-specific antigen, Gleason score, surgical margin status, extracapsular extension, seminal vesicles invasion and lymph node involvement (HR = 2.46, 95% CI 1.34-4.52; $P = 0.004$). Altogether, these analyses provide compelling evidence that patients harboring a tumor with concurrent AR_low/MYC_high transcriptional signatures are more likely to develop a metastatic disease compared to AR_low/MYC_low patients.

Is the functional role of MYC cell type-specific? if so, please comment that in the text when talking about AP, DLP, VP lobes and other regions of prostate.

Response: This is an interesting question. We have performed Gene Sets Enrichment Analysis (GSEA; Hallmark) comparing luminal ($Krt8^{Hi}$, $Krt18^{Hi}$) MYC transgene overexpressing cell subpopulation (Hi-MYC model) to the luminal WT subpopulation (WT animal). While key transcriptional programs driven by MYC overexpression were found to be similar between cells of VP and DLP origin (e.g. MYC_targets_V1/V2; Androgen_response), the AP lacked these hallmark features (**Reviewer Figure 2A**). However, since only very few luminal cells expressed the MYC transgene in the AP (17 cells) compared to the DLP (75 cells) or the VP (976 cells), this discrepancy could be due to a limited and incomplete transcriptional profiling rather than a genuine lobe-specific difference in the functional role of MYC. Therefore, we sampled down both the VP and DLP to 17 cells and performed GSEA. Critically, our results revealed that MYC_targets_V1/V2 were no longer enriched in the VP and DLP, suggesting that our current sampling of MYC-overexpressing cells in the AP is not sufficient for the assessment of MYC-driven transcriptional programs (**Reviewer Figure 2B**). Taken altogether, our data do not suggest a cell type or lobe specific functional role for MYC. However, a greater sampling of luminal ($Krt8^{Hi}$, $Krt18^{Hi}$) MYC transgene overexpressing cells in the AP and the DLP would be required to provide a definitive answer to this question.

Reviewer Figure 2: (A) Gene Set Enrichment Analysis (GSEA; Hallmark) revealed that luminal MYC transgene overexpressing cells are associated to an enriched MYC transcriptional activity (MYC_targets_V1/V2) and depleted Androgen_response (compared to the luminal WT subpopulation from WT animal) in the VP and DLP but not in the AP. **(B)** Sampled down GSEA (Hallmark) performed on 17 randomly selected luminal MYC transgene overexpressing cells is no longer able to capture enriched MYC transcriptional activity in the VP and DLP. ($P < 0.05$ and $FDR < 0.1$)

Line 130: Perhaps this is just semantics but I do not see "a robust and uniform MYC-driven PIN transition" in DLP case, presented in Figure 11.

Response: The sentence that the reviewer is referring to *only* mention the VP but *does not* mention the DLP (Line 130: *The high representation of luminal cells coupled with a robust and uniform MYC-driven PIN transition in the VP enabled us to further define distinct luminal subpopulations.*)

Lines 145-147. I find following claim confusing: "a state that was independent of human or murine MYC transcript level (Figure 2D)". Looking at Figure 2D it seems that there are differences in expression (when comparing Lum1/2 vs LumMYC/LumMYC high, distributions).

Response: Unlike bulk RNA-seq, scRNA-seq generates a large number of samples (*i.e.* cells) for each group we are comparing. Thus, we can take advantage of the whole distribution of expression values in each group to identify differences between groups rather than comparing estimates of mean expression as it is standard for bulk RNA-seq. There are two main approaches to comparing distributions.

Firstly, we can use existing statistical models/distributions and fit the same type of model to the expression in each group then test for differences in the parameters for each model, or test whether the model fits better if a particular parameter is allowed to be different according to group. For instance, we used edgeR in our manuscript to test whether allowing mean expression to be different in different batches significantly improved the fit of a negative binomial model of the data.

Alternatively, we can use a nonparametric test which does not assume that expression values follow any particular distribution (*e.g.* Wilcoxon rank-sum test). Nonparametric tests generally convert observed expression values to ranks and test whether the distribution of ranks for one group are significantly different from the distribution of ranks for the other group (<https://scrnaseq-course.cog.sanger.ac.uk/website/biological-analysis.html#bulk-rna-seq-1>).

Seurat is a commonly used tool to identify differentially expressed genes from scRNA-seq data ¹², the default of Seurat is Wilcoxon rank-sum test.

Importantly, using both edgeR and Seurat, we confirmed that the expression of murine MYC (mm10MYC) is not affected by human MYC (hg19MYC) overexpression and that the overexpression of human MYC is not different between Luminal_MYC subpopulations (**Reviewer Table 1**).

Reviewer Table 1: Expression of murine MYC (mm10MYC) is not altered following human MYC overexpression (hg19MYC) and human MYC overexpression is not different between Luminal_MYC subpopulations.

Differential Gene Expression Method	Luminal_MYC over Luminal_WT (mm10MYC)			Luminal_MYC (high prolifer.) over Luminal_MYC (hg19MYC)		
	Log2FC	P-value	FDR	Log2FC	P-value	FDR
edgeR	0.01719073	0.779880067	1	-0.07119074	1	1
Seurat		0.001697287	1		0.0669928	1

Line 150: "Taken together, these results demonstrate that MYC-driven transcriptional reprogramming can be readily captured by single-cell transcriptomics". I think such conclusion is just too obvious. scRNA-Seq has advances far beyond this point.

Response: We agree with the reviewer and we have modified the sentence as follows: "Taken together, these results demonstrate that MYC-driven transcriptional reprogramming can be readily captured *in vivo* by single-cell transcriptomics to expose inter- and intra-prostate lobe heterogeneity."

Line 182: "... these results indicate that AR-transcriptional program is compromised upon MYC overexpression." Again, perhaps this is just semantics, but if MYC overexpression suppresses or outcompetes the AR-activated genes, because of shared cis elements, it would be incorrect to claim that "AR-transcriptional program is compromised".

Response: The reanalysis of the Barfeld *et al.* ChIP-seq data revealed that AR binding nearby Androgen_response genes remained largely unchanged following MYC overexpression⁹. Importantly, MYC binding nearby MYC_targets_V1 genes also remained unchanged following MYC overexpression despite a significant enrichment of the MYC_targets_V1 gene set (**Supplementary Figure S9C**). Additionally, AR binding was found to be increased at genomic regions nearby Androgen_response genes alongside the H3K27ac mark following MYC overexpression in the Hi-MYC model (**Figure 7B-C**), in stark contrast with the accompanied depletion of the Androgen_response gene set (**Figure 3C**). Therefore, our data ***does not*** suggest that MYC overexpression suppresses or outcompetes the AR-activated genes because of shared cis elements. Rather our results ***support cofactor redistribution*** driven by increased MYC expression and resulting in greater RNA Pol II promoter-proximal pausing as a potential mechanism for MYC-mediated transcriptional repression at genes regulated specifically by the AR. This hypothesis is currently investigated by our laboratory.

Material and Methods

The dissociation protocol seems to be very long (>2h hours), a time during which cells are expected to experience stress and change their transcriptional outcome drastically. Investigating subtle effects such as transcriptional pausing on sample that were dissociated for 2 hours raises questions about introduction of technical biases. I recognized that my request might be stretched too far but to rule out biological and technical artifacts I think you would need alternative confirmation that transcriptional pausing is actually happening, for example on cells that are not being dissociated for such a long time (e.g. using nuc-seq).

Response: We would like to clarify our experimental workflow and reassure the reviewer. As mentioned in the Methods section (Line 518): "Mouse prostate lobes (AP, DLP, VP) were dissected, weighed and immediately processed for bulk and single-cell transcriptomics or flash-frozen in liquid nitrogen for chromatin immunoprecipitation or protein expression experiments." Therefore, all our ChIP-seq experiments, including the RNA Pol II ChIP-seq used to quantify transcriptional pause release at AR-regulated genes, used flash-frozen murine prostate lobes as starting material, ruling out potential biological and technical artifacts that could have been associated with a dissociation protocol.

Reviewer #3

Expert in AR signalling and prostate cancer

Remarks to the Author

The manuscript by Qiu et al. describes the crosstalk between MYC expression androgen receptor signaling in the mouse prostate. Authors compared the transcriptomes single cells derived from lobes of the three prostate lobes, the anterior, dorsolateral and ventral prostate. In addition, authors compared the transcriptomes of the lobes of probasin-MYC transgenic mice driving an overexpression of the MYC oncogene in the prostate leading to PIN. The data suggest a repression of androgen receptor (AR) signaling in the lobes affected by MYC overexpression without reducing AR gene expression itself. Furthermore, ChIP-seq with primary tissues were performed, integrating AR, FOXA1 and H3K27ac ChIP-seq to single-cell transcriptomes. TCGA datasets indicates a correlation among human PCa characterized by a low AR transcriptional signature with concurrent high MYC transcriptional signature to be associated with a short time to biochemical recurrence whereas high AR in combination with low MYC signature associates with the longest time to biochemical recurrence. Gene expression profiling from 59 AR-positive CRPC tumors revealed that AR activity is negatively correlated with MYC expression supporting that MYC inhibits the canonical AR transcriptional program in CRPC.

LuCaP patient-derived xenografts were also used to analyze cistrome and FOXO1 occupancy. A greater FOXA1 occupancy was observed at AR gained binding sites in MYC-high compared to low MYC. Functionally these observations were confirmed by overexpressing MYC in the mouse model system.

AR binding was significantly associated with genes downregulated by MYC overexpression. However interestingly, AR binding without inactivation of chromatin nearby androgen response genes remained largely unchanged following MYC overexpression as well as MYC binding nearby MYC target genes also remained unchanged by MYC overexpression despite MYC inhibits AR transcriptome signature. Authors suggest mechanistically, shown by ChIP-seq of Pol II, that Pol II pauses at promoters upon MYC overexpression.

Since these patterns suggest a MYC driven altered ratio of initiating and elongating RNA Pol II at AR-regulated genes.

Authors have impressively shown in native tissue the crosstalk between MYC and AR in in vivo mouse model and human PCa cell derived xenografts. Furthermore, the mechanism suggested is that MYC overexpression leads to pausing of Pol II in a subset of androgen responsive genes.

Response: We thank the reviewer for the encouraging comments.

Minor points:

- *Please indicate how many fresh prostate lobes were used in the bulk RNA seq?*

Response: Each prostate lobe was dissociated to form a single cell suspension that was then divided for bulk and single-cell RNA-sequencing. We have clarified the Methods section accordingly.

- On Page 14 it is mentioned tyroptinization. In which step that enzyme was used?

Response: Trypsin was used after the collagenase/hyaluronidase dissociation step. Our cell dissociation procedure was optimized to generate viable single cell suspension from fresh murine prostate tissues.

- Fig. 4H and Fig. 7K: GSEA analysis indicates that the p53 pathway is involved in both MYC dependent increase and decrease.

Response: This is an interesting observation. We have dissected the genes represented in p53_pathway (**Figure 4H**). Importantly, p53_pathway genes associated with a MYC-dependent increase (Cluster 1) were not found in the list of p53_pathway genes associated with a MYC-dependent decrease (Cluster 2) (**Reviewer Figure 3A**). We performed the same exercise with p53_pathway genes associated with RNA Pol II pause release (**Figure 7K**) and RNA Pol II pause (**Figure 7M**) following MYC overexpression. Again, there was no overlap between both lists of p53_pathway associated genes (**Reviewer Figure 3B**). Therefore, our data support the notion that a subset of p53_pathway genes are upregulated while another, non-overlapping subset of p53_pathway genes, are downregulated following MYC overexpression.

Reviewer Figure 3: List of p53_pathway genes associated with (A) MYC-dependent increase or decrease (related to **Figure 4H**) and (B) RNA Pol II pause release or pause (related to **Figure 7K, M**).

Although this is bioinformatically possible, the request would be to analyze the same data in addition with an IPA (Ingenuity Pathways Analysis).

Response: As suggested, we have performed Ingenuity Pathways Analysis (IPA) to gain additional insights on the function of genes that are associated with RNA Pol II pause release (**Figure 7I, J, K**) and RNA Pol II pause (**Figure 7I, L, M**) following MYC overexpression. Interestingly, IPA canonical pathways associated with RNA Pol II pause release includes mitogenic/growth (EIF2 signaling, mTOR signaling, Regulation of eIF4 and p70S6K signaling, Cell cycle: G1/S checkpoint regulation) and MYC-related (MYC Mediated Apoptosis Signaling) pathways (**Reviewer Figure 4A**). Critically, IPA canonical pathways associated with RNA Pol II pause include the assembly of RNA Pol II complex and androgen signaling (**Reviewer Figure 4B**). Taken altogether, IPA analysis provides an orthogonal confirmation that RNA Pol II pause release is associated with MYC-related processes while RNA Pol II pause with a depletion of AR transcriptional activity following MYC overexpression.

Reviewer Figure 4: Top ten Ingenuity Pathway Analysis (IPA) canonical pathways associated with (A) RNA Pol II pause release (related to **Figure 7I, J, K**) or (B) RNA Pol II pause (related to **Figure 7I, L, M**).

Reviewer #4

Expert in MYC signalling

Remarks to the Author

Qiu *et al.*,

Qiu *et al.* describe that MYC overexpression in prostate cancer cells hinders the release of paused RNA Pol2 at androgen receptor (AR) target genes, thereby reducing the transcriptional output of AR-driven transcriptional programs. This transcriptional shift leads to a more aggressive phenotype of prostate cancer cells. While most of the data support this hypothesis and are scientifically sound, the data structure of the manuscript and its presentation in the figures is partially confusing and to some extent hinders a clear understanding of the principal findings. Deconvolution of the data reveals limited novelty in comparison to the findings of Barfeld *et al.*, (PMID 28412251) already demonstrating the AR-antagonizing role of MYC in prostate cancer. In addition, there is apparently some controversy in the field with respect to the role of MYC as there is no effect on AR as shown in a mouse model (Kim *et al.* *Oncogene* 2012) and in cell lines models AR knockdown has been shown to reduce the expression of MYC (Gao *et al.* *Plos One* 2013). The present study mechanistically expands this observation by demonstrating MYC-induced hinderance of RNA Pol2 pause release at AR target genes. To increase the novelty of the study, ideally, this finding should be followed up mechanistically in more detail to justify publication in this journal. As is, due to lack of novelty, and limited functional and translational follow-up the reviewer cannot recommend publication of the manuscript in its present form.

Response: We have contextualized the novelty of our work with regards to the Barfeld *et al.* manuscript (REF 15)⁹ in our response to Reviewer #2. Regarding the impact of AR expression / activation on the expression of MYC as described by Gao *et al.*¹³, this is a topic that was investigated in depth by another research team in a manuscript currently under consideration at *Nature Communications* (Guo, Yimming *et al.*). In the Kim *et al.* manuscript, which uses the Z-MYC genetically engineered mouse model (GEMM) of prostate cancer, authors ***did not*** investigate the c-MYC driven transcriptional reprogramming in murine prostates¹⁴. Importantly, authors demonstrate via immunohistochemistry stainings that the AR remain expressed in all their GEMM (*i.e.* c-MYC⁺;Pten^{+/-};p53^{+/-} ; c-MYC⁺;Pten^{+/-};p53^{-/-} ; c-MYC⁺;Pten^{-/-};p53^{+/-}). This finding ***does not*** contradict our results since we also found that AR expression both at the transcript and protein level remain unchanged following c-MYC overexpression (compared to wild-type animals; **Figure 3D-E**). Instead, our manuscript identifies a novel mechanism linking MYC and AR that have a profound influence on activities of both, on prostate cancer development, and on response to therapy.

General comments:

1. For increased clarity Figures 1, 2 and 3 should be combined into one Figure that is condensed to principal findings of MYC activation influencing transcription in the mouse model by diminishing AR transcriptional output. Especially Figure 1B/C/D, Figure 2 and the second part of Figure 3 are not very intuitive and add little to the overall finding.

Response: To our knowledge, this is the first time that the Hi-MYC genetically engineered mouse model (GEMM) transcriptome of the AP, VP and DLP lobes have been characterized at a single cell level along with wild-type prostates. Thus, we believe that **Figure 1** and **Figure 2** will be of great interest to the prostate cancer scientific community. Regarding the covariance analysis in **Figure 3F-G**, we believe this represents a powerful application of single-cell RNA-sequencing in defining transcriptional programs at a single-cell level. Therefore, we would prefer **keeping Figures 1, 2 and 3** unaltered.

2. Mechanistically, it would be interesting to answer whether increased MYC binding at AR target sites is dependent on MYC binding to canonical E-Box sequence motifs or not. This is interesting to understand MYC-induced gene repression and whether it differs from MYC-induced gene induction on a DNA binding sequence level.

Response: The reanalysis of the Barfeld *et al.* ChIP-seq data revealed that MYC binding nearby Androgen_response genes remained largely unchanged following MYC overexpression⁹. Moreover, MYC binding nearby MYC_targets_V1 genes also remained unchanged following MYC overexpression despite a significant enrichment of the MYC_targets_V1 gene set (**Supplementary Figure S9C**). Therefore, our data does not suggest that MYC-induced gene induction or MYC-induced gene repression rely on the redistribution of MYC binding to *cis* elements (canonical E-Box sequence motifs or not). Rather, as discuss in our response to Reviewer #2, our results support cofactor redistribution driven by increased MYC expression and resulting in greater RNA Pol II promoter-proximal pausing as a potential mechanism for MYC-mediated transcriptional repression at genes regulated specifically by the AR. Regardless, we reanalyzed the Barfeld *et al.* MYC ChIP-seq data to determine whether MYC binding nearby Androgen_response genes is dependent of canonical E-Box sequence motifs or not. Interestingly, MYC binding nearby Androgen_response genes was associated to E-Box sequence motifs (E-box(bHLH)/Promoter/Homer: $P = 1e-13$) to a similar extent as MYC binding at randomly selected sites not related to Androgen_response genes (E-box(bHLH)/Promoter/Homer: $P = 1e-11$). This support that MYC-induced gene repression does not differ from MYC-induced gene induction on a DNA binding sequence level.

3. *What is the translational impact of these results? The authors should present data that would hint at a possible therapeutic intervention that would harness the findings either in the utilized mouse model or in cellular models. This type of data may justify the publication in this journal.*

Response: This is an important question. Along this line, Bai *et al.* recently showed that a c-Myc inhibitor disrupting c-Myc and Max dimerization sensitizes enzalutamide-resistant prostate cancer cells to growth inhibition by enzalutamide¹⁵, suggesting that c-MYC is not only key to prostate cancer etiology, but also to resistance to standard-of-care therapies. Based on the hypothesis that c-MYC transcriptional activity is central to the response to next generation AR signaling inhibitor (ARSI; abiraterone or enzalutamide) treatment, we used gene expression data to stratify 75 metastatic castration-resistant prostate cancer (mCRPC) in the SU2C International Dream Team cohort dataset based on the combined levels of the Hallmark Androgen_response (high; low) and MYC_targets_V1 (high; low) transcriptional signatures¹⁶. Strikingly, Kaplan-Meier curves and univariable analysis revealed that patients with mCRPC tumors harboring an AR_low/MYC_high signature were more likely to resist ARSI treatment and die of their disease (Hazard Ratio (HR) = 9.75, 95% Confidence Interval (CI) 3.02-31.55; $P < 0.001$; **Reviewer Figure 5A-B**). These additional analyses suggest that concurrent AR_low/MYC_high transcriptional signatures identify a subgroup of patients that are predisposed to fail standard-of-care therapies and progress to develop metastatic disease (**Figure 5**) but also to fail first-line next generation ARSI treatment and die of mCRPC (**Reviewer Figure 5A-B** and incorporated to **Figure 6**). Altogether, these results support the use of therapies not centered on the inhibition of AR signaling (e.g. PARP inhibitors, [¹⁷⁷Lu]Lu-PSMA-617) for the subgroup of patients harboring concurrent AR_low/MYC_high transcriptional programs. This is now mentioned in the discussion section of our manuscript.

Reviewer Figure 5: Divergent MYC and AR transcriptional programs dictate response to first-line AR signaling inhibitor (ARSI). (A, B) Kaplan Meier curves (A) and univariable analysis (B) revealed that patients bearing a metastatic castration-resistant prostate cancer (mCRPC) characterized by an AR_low/MYC_high signature have shorter time to death from the start of first line ARSI (abiraterone or enzalutamide) within the SU2C International Dream Team cohort (Abida *et al.*), 2019.

Minor points:

1. *Fig 1D/E/F the authors should comment / offer an explanation why high MYC does not alter transcription in the AP lobe.*

Response: This is an interesting question that has been answered in our response to Reviewer #1. Briefly, the marginal prostate intraepithelial neoplasia (PIN) penetrance observed in the AP lobe is most likely due to the weak expression of the human *MYC* transgene in this lobe (**Figure 1I**) as the level of murine prostate transformation was shown to be proportional to the strength of the human *MYC* transgene expression ¹.

2. *Fig 1H/I the authors claim PIN transformation is MYC-driven. However, PIN transformation in the DLP is comparable to VP (Fig1H - no statistical difference indicated) despite lower MYC levels and 10x lower number of luminal cells in the DLP vs VP. This contradiction must be explained. Otherwise, the claim that PIN is MYC-driven cannot be made.*

Response: Again, this is an interesting question that has been raised by Reviewer #1 and answered above. Briefly, in the Hi-MYC GEMM of prostate cancer, levels of MYC overexpression in the DLP are sufficient in leading to comparable PIN penetrance to the VP of 12-week-old animals. However, the kinetic of the DLP transformation is likely slower than for the VP due to weaker MYC transgene overexpression as it was previously demonstrated in the Lo-MYC GEMM of prostate cancer ¹.

3. *Fig S4 the figure is hardly comprehensible due to the very poor contrast and expression differences are not clearly visible – this must be displayed more clearly.*

Response: We agree with the Reviewer and apologize for the poor quality of the figure. We have increased the size of each cell represented on the tSNE plots between 200% (**Figure 1, Figure 2, Figure S3**) and 400% (**Figure S2, Figure S4, Figure S5**) to improve readability.

4. *Fig 2B/3A: The two Luminal MYC subpopulations do not differ in MYC expression, however, only a minority is “highly proliferating”. It would be interesting to include this distinction in subpopulation also in the gene set analysis in Fig 3A and evaluate the contribution of either luminal subpopulation to the overall luminal transcriptional profile. Given the possible different origin of these subpopulations and comparable MYC expression but differential aggressiveness, it would be interesting to evaluate how and why comparable MYC levels lead to differential transcriptional output.*

Response: We thank the reviewer for this interesting question. We have performed Gene Set Enrichment Analysis (GSEA) comparing Luminal_MYC (high prolifer.) over Luminal_MYC. Only four gene sets were significantly enriched ($P < 0.05$ and $FDR < 0.1$), three of which are directly related to cell proliferation (G2M_checkpoint, E2F_targets, Mitotic_spindle) and none were depleted (**Reviewer Figure 5A**). Accordingly, assessment of the cell-cycle phase distribution using the Seurat Cell-Cycle Scoring and Regression package across luminal (WT and MYC) subpopulations revealed that all Luminal_MYC (high prolifer.) cells are either in the S (54%) or G2M (46%) phase (**Reviewer Table 2**). This was associated with greater levels in the expression of proliferation markers (*e.g.* *Top2a*, *Mki67*, *Pcna*; **Reviewer Figure 5B**). Altogether, our results suggest that albeit differences in proliferation-related pathways/genes, MYC transcriptional program is similar between Luminal_MYC (high prolifer.) and Luminal_MYC subpopulations.

Reviewer Figure 6: Luminal_MYC (high prolifer.) subpopulation is enriched for proliferation-related pathways/genes. (A) Gene Set Enrichment Analysis (GSEA, Hallmark, $P < 0.05$ and $FDR < 0.1$) revealed that Luminal_MYC (high prolifer.) transcriptional program is enriched for proliferation-related gene sets (compared to the Luminal_MYC subpopulation). (B) Expression level of selected proliferation-related markers reveals enrichment in the Luminal_MYC (high prolifer.) subpopulation.

Reviewer Table 2: Cell-cycle phase distribution of the luminal cell subpopulations defined by the Seurat Cell-Cycle Scoring and Regression package.

Subpopulations	Cell (number)			Cell (percentage)		
	G1	S	G2M	G1	S	G2M
Luminal_1_WT	387	84	5	81	18	1
Luminal_2_WT	447	78	16	83	14	3
Luminal_MYC	779	377	34	65	32	3
Luminal_MYC (high prolifer)	0	53	46	0	54	46

REFERENCES

1. Ellwood-Yen K, *et al.* Myc-driven murine prostate cancer shares molecular features with human prostate tumors. *Cancer Cell* **4**, 223-238 (2003).
2. Parisotto M, Metzger D. Genetically engineered mouse models of prostate cancer. *Mol Oncol* **7**, 190-205 (2013).
3. Bostwick DG, Liu L, Brawer MK, Qian J. High-grade prostatic intraepithelial neoplasia. *Rev Urol* **6**, 171-179 (2004).
4. Lin CY, *et al.* Transcriptional amplification in tumor cells with elevated c-Myc. *Cell* **151**, 56-67 (2012).
5. Sabo A, *et al.* Selective transcriptional regulation by Myc in cellular growth control and lymphomagenesis. *Nature* **511**, 488-492 (2014).
6. Walz S, *et al.* Activation and repression by oncogenic MYC shape tumour-specific gene expression profiles. *Nature* **511**, 483-487 (2014).
7. Muhar M, *et al.* SLAM-seq defines direct gene-regulatory functions of the BRD4-MYC axis. *Science* **360**, 800-805 (2018).
8. Chen FX, Smith ER, Shilatifard A. Born to run: control of transcription elongation by RNA polymerase II. *Nat Rev Mol Cell Biol* **19**, 464-478 (2018).
9. Barfeld SJ, *et al.* c-Myc Antagonises the Transcriptional Activity of the Androgen Receptor in Prostate Cancer Affecting Key Gene Networks. *EBioMedicine* **18**, 83-93 (2017).
10. Street K, *et al.* Slingshot: cell lineage and pseudotime inference for single-cell transcriptomics. *BMC Genomics* **19**, 477 (2018).
11. Saelens W, Cannoodt R, Todorov H, Saeys Y. A comparison of single-cell trajectory inference methods. *Nat Biotechnol* **37**, 547-554 (2019).
12. Stuart T, *et al.* Comprehensive Integration of Single-Cell Data. *Cell* **177**, 1888-1902 e1821 (2019).
13. Gao L, *et al.* Androgen receptor promotes ligand-independent prostate cancer progression through c-Myc upregulation. *PLoS One* **8**, e63563 (2013).
14. Kim J, Roh M, Doubinskaia I, Algarroba GN, Eltoum IE, Abdulkadir SA. A mouse model of heterogeneous, c-MYC-initiated prostate cancer with loss of Pten and p53. *Oncogene* **31**, 322-332 (2012).
15. Bai S, *et al.* A positive role of c-Myc in regulating androgen receptor and its splice variants in prostate cancer. *Oncogene* **38**, 4977-4989 (2019).
16. Abida W, *et al.* Genomic correlates of clinical outcome in advanced prostate cancer. *Proc Natl Acad Sci U S A* **116**, 11428-11436 (2019).

Reviewers' Comments:

Reviewer #1:

Remarks to the Author:

The authors have addressed all my concerns except for point 5 regarding probasin western blot analyses (Figure 7E). The authors argued that "The upper band is most likely unspecific to Probasin", but could they perform a knockdown assay to support their conclusion? Additionally, the loading control levels were not comparable between WT and MYC.

Reviewer #2:

Remarks to the Author:

Thank you for addressing the reviewer's comments and critique. I have no more comments or suggestions that could further improve this work and therefore support its publication.

Reviewer #3:

None

Reviewer #4:

Remarks to the Author:

Qiu and Boufaied et al., have sufficiently addressed most of the raised questions, which solidifies their findings and strengthens their manuscript. However, they did not provide experimental data to address the translational impact of their results (Major point #3 of the review concerns) even though their provided Kaplan-Meier analysis led them to conclude:

'These additional analyses suggest that concurrent AR_low/MYC_high transcriptional signatures identify a subgroup of patients that are predisposed to fail standard-of-care therapies and progress to develop metastatic disease (Figure 5) but also to fail first-line next generation ARSI treatment and die of mCRPC (Reviewer Figure 5A-B and incorporated to Figure 6). Altogether, these results support the use of therapies not centered on the inhibition of AR signaling (e.g. PARP inhibitors, [177Lu]Lu-PSMA-617) for the subgroup of patients harboring concurrent AR_low/MYC_high transcriptional programs.'

This conclusion states a very clear hypothesis that could be experimentally tested in cellular models: whether 'therapies not centered on the inhibition of AR signaling' like PARP inhibitors are efficacious in models with AR_low/MYC_high transcriptional programs?

We would like to thank all reviewers in helping improve our manuscript through insightful comments and discussions.

REVIEWER COMMENTS

Reviewer #1

Expert in ChIP-seq and RNA-seq analysis

Remarks to the Author

The authors have addressed all my concerns except for point 5 regarding probasin western blot analyses (Figure 7E). The authors argued that “The upper band is most likely unspecific to Probasin”, but could they perform a knockdown assay to support their conclusion? Additionally, the loading control levels were not comparable between WT and MYC.

Response: In order to thoroughly answer this concern, we purchased murine Probasin ORF clone from Origene (#MR2152286) and performed a transient transfection in MyC-CaP murine prostate cancer cells (negative for Probasin expression). As expected, Probasin transcript levels were significantly increased following transfection (**Reviewer Figure 1A**). However, the Probasin antibody (#sc-393830, Santa Cruz Biotechnology) failed to detect protein expression in the *Probasin* transfected cells by western blot (**Reviewer Figure 1B**). Surprisingly, the Probasin antibody revealed the same band pattern in the anterior prostate (AP) and dorsolateral prostate (DLP) as in the ventral prostate (VP), whereas the lower band at the expected Probasin molecular weight (~22 kDa) is lost in MYC-expressing tissues (**Reviewer Figure 1B**). Although this is in accordance with our scRNA-seq data (**Figures 3G, 7E**) and our RNA Pol II ChIP-seq (**Figure 7M**), this experiment raises important doubts on the specificity of this Probasin antibody. Therefore, we decided to remove the Probasin western blot from our manuscript (previously **Figure 7E** and **Supplementary Data 2**). Importantly, this does not alter the conclusions of our manuscript in which MYC overexpression antagonizes the canonical AR transcriptional program by disrupting transcriptional pause release at AR-regulated genes.

Reviewer Figure 1: Probasin antibody from Santa Cruz Biotechnology (#sc-393830) is not specific. (A) Murine MyC-CaP prostate cancer cells express high levels of Probasin transcript following transient transfection (mean \pm SD). **(B)** Probasin antibody from Santa Cruz Biotechnology (#sc-393830) fail to detect Probasin in MyC-CaP overexpressing cells although it detects a band in WT prostate tissues at the expected molecular weight (~22 kDa). EV: empty vector; WT: wild-type; AP: anterior prostate; DLP: dorsolateral prostate.

Reviewer #2

Expert in sc-RNAseq analysis

Remarks to the Author

Thank you for addressing the reviewer's comments and critique. I have no more comments or suggestions that could further improve this work and therefore support its publication.

Response: Thank you for your help in improving our manuscript.

Reviewer #3

Expert in AR signalling and prostate cancer

Remarks to the Author

Response: We would like to thank the reviewer again for the encouraging comments.

Reviewer #4

Expert in MYC signalling

Remarks to the Author

Qiu and Boufaied et al., have sufficiently addressed most of the raised questions, which solidifies their findings and strengthens their manuscript. However, they did not provide experimental data to address the translational impact of their results (Major point #3 of the review concerns) even though their provided Kaplan-Meier analysis led them to conclude:

'These additional analyses suggest that concurrent AR_low/MYC_high transcriptional signatures identify a subgroup of patients that are predisposed to fail standard-of-care therapies and progress to develop metastatic disease (Figure 5) but also to fail first-line next generation ARSI treatment and die of mCRPC (Reviewer Figure 5A-B and incorporated to Figure 6). Altogether, these results support the use of therapies not centered on the inhibition of AR signaling (e.g., PARP inhibitors, [177Lu]Lu-PSMA-617) for the subgroup of patients harboring concurrent AR_low/MYC_high transcriptional programs.'

This conclusion states a very clear hypothesis that could be experimentally tested in cellular models: whether 'therapies not centered on the inhibition of AR signaling' like PARP inhibitors are efficacious in models with AR_low/MYC_high transcriptional programs?

Response: In our initial rebuttal, this reviewer asked an important question whereby "The authors should present data that would hint at a possible therapeutic intervention that would harness the findings either in the utilized mouse model or in cellular models." Along this line, we highlighted that Bai *et al.* recently showed that a c-Myc inhibitor disrupting c-Myc and Max dimerization sensitizes enzalutamide-resistant prostate cancer cells to growth inhibition by enzalutamide *in vitro*¹, suggesting that c-MYC is not only key to prostate cancer etiology, but also to resistance to standard-of-care therapies. Based on these results and the novel findings presented in our manuscript (e.g., an active MYC transcriptional program and low AR activity identify prostate cancer patients predisposed to fail standard-of-care therapies and most likely to develop metastatic castration resistant prostate cancer (mCRPC), we provided further evidence that

AR_{low}/MYC_{high} transcriptional signatures also identify a subgroup of patients that are predisposed to fail first-line next generation AR signaling inhibitors (ARSI; abiraterone or enzalutamide) treatment and die of mCRPC (**Figure 6J, K**). Therefore, we discuss this later finding in our manuscript and mention that the use of therapies not centered on the inhibition of AR signaling (*e.g.*, PARP inhibitors, [¹⁷⁷Lu]Lu-PSMA-617) for the subgroup of patients harboring concurrent AR_{low}/MYC_{high} transcriptional programs should be considered for mCRPC patients since they do not respond to ARSI treatment (**Figure 6J, K**).

In May 2020, the Food and Drug Administration (FDA) approved the use of PARP inhibitors Olaparib and Rucaparib for men with mCRPC disease that progressed following prior treatment with ARSI based on the PROFOUND and TRITON2 trials, respectively ^{2,3}. While these therapies have been approved for mCRPC in patients with tumors that harbor germline or somatic mutations in DNA damage response (DDR) genes (*i.e.*, ~13-20% of patients), the exact mechanism dictating response to PARPi remains incompletely understood as some patients without germline or somatic mutations in DDR genes also show response to therapy ⁴. Similarly, based on the VISION study ⁵, the FDA granted priority review in September 2021 to the targeted radioligand therapy [¹⁷⁷Lu]Lu-PSMA-617 to treat patients with mCRPC that progressed following prior treatment with ARSI and who had PSMA-positive [⁶⁸Ga]Ga-PSMA-11 positron-emission tomographic-computed tomographic scans.

Considering that mCRPC patients bearing a tumor with an AR_{low}/MYC_{high} transcriptional signatures do not respond to ARSI treatment (**Figure 6J, K**), we simply suggest that alternative treatment modalities, which demonstrated efficacy for men with mCRPC disease, should be considered first instead of ARSI treatment for this subset of patients. We agree with this reviewer that defining the best treatment strategy for AR_{low}/MYC_{high} mCRPC tumors is of great interest. While we are also very enthusiasts by the potential clinical implications of our findings, this endeavor falls outside of the scope of our manuscript that focuses on the interplay between MYC and AR transcriptional programs. However, we are currently working toward that goal with a combination of *in vitro* and *in vivo* models as well as clinical data that will require an independent manuscript in order to be adequately presented.

REFERENCES

- 1 Bai, S. *et al.* A positive role of c-Myc in regulating androgen receptor and its splice variants in prostate cancer. *Oncogene* **38**, 4977-4989, doi:10.1038/s41388-019-0768-8 (2019).
- 2 de Bono, J. *et al.* Olaparib for Metastatic Castration-Resistant Prostate Cancer. *N Engl J Med* **382**, 2091-2102, doi:10.1056/NEJMoa1911440 (2020).
- 3 Abida, W. *et al.* Rucaparib in Men With Metastatic Castration-Resistant Prostate Cancer Harboring a BRCA1 or BRCA2 Gene Alteration. *J Clin Oncol* **38**, 3763-3772, doi:10.1200/JCO.20.01035 (2020).
- 4 Mateo, J. *et al.* DNA-Repair Defects and Olaparib in Metastatic Prostate Cancer. *N Engl J Med* **373**, 1697-1708, doi:10.1056/NEJMoa1506859 (2015).
- 5 Sartor, O. *et al.* Lutetium-177-PSMA-617 for Metastatic Castration-Resistant Prostate Cancer. *N Engl J Med* **385**, 1091-1103, doi:10.1056/NEJMoa2107322 (2021).